# Encapsulated CdSe/CdS nanorods in double-shelled porous nanocomposites for efficient photocatalytic $CO_2$ reduction

Hui Li[1], Caikun Cheng[1], Zhijie Yang[1] & Jingjing Wei [1] ✉

Colloidal quantum dots have been emerging as promising photocatalysts to convert $CO_2$ into fuels by using solar energy. However, the above photocatalysts usually suffer from low $CO_2$ adsorption capacity because of their nonporous structures, which principally reduces their catalytic efficiency. Here, we show that synchronizing imine polycondensation reaction to self-assembly of colloidal CdSe/CdS nanorods can produce micro-meso hierarchically porous nanocomposites with double-shelled nanocomposites. Owing to their hierarchical pores and the ability to separate photoexcited electrons, the self-assembled porous nanocomposites exhibit remarkably higher activity ($\approx$ 64.6 μmol g$^{-1}$ h$^{-1}$) toward $CO_2$ to CO in solid-gas regime than that of nonporous solids from self-assembled CdSe/CdS nanorods under identical conditions. Importantly, the length of the nanorods is demonstrated to be crucial to correlate their ability to long-distance separation of photogenerated electrons and holes along their axial direction. Overall, this approach provides a rational strategy to optimize the $CO_2$ adsorption and conversion by integrating the inorganic and organic semiconductors.

Nature created sophisticated inorganic/organic systems to store solar energy into chemical bonds in an efficient yet long-lasting way[1]. Chemists have long been inspired by biological systems in their efforts to create artificial photosynthetic systems, which potentially lead to efficient solar-to-fuel conversion and achieve a sustainable carbon cycle[2–4]. In particular, the photoreduction of $CO_2$, a main category of greenhouse gases, into fuels (e.g. CO or hydrocarbons) could potentially address the energy and environmental issues in a clean yet sustainable way[5,6]. Compared with the homogeneous process catalyzed by molecular complexes, the heterogeneous process might be more intriguing for the photocatalytic $CO_2$ reduction, where the reduction of $CO_2$ directly takes place at the interface between the catalysts and the dispersing medium either in liquid or gas, leading to the distinct solid-liquid and solid-gas modes[7–9]. Although remarkable progress has been made for the $CO_2$ photoreduction in solid-liquid mode, they remain suffer from the low catalytic activity because of the limited solubility of $CO_2$ in the liquid and the difficulty in the separation of the products when they could be dissolved in the liquid[10,11]. In sharp contrast, photoreduction at the solid-gas interface, where solid catalysts are deposited on a support and are exposure to the reactive gases comprised of $CO_2$ and $H_2O$ (provides protons and electrons), would overcome these limitations[12,13]. Nevertheless, grand challenges remain in exploring highly efficient photocatalytic systems in a solid-gas regime, which require porosity engineering to have high $CO_2$ adsorption capacity[14] and band-structure engineering that enables strong incident light absorption and efficient charge separation[15].

Searching for diverse materials, semiconducting quantum dots (QDs), in particular those chalcogenides-based II-IV semiconductors, are emerging as ideal candidates in the field of photocatalytic $CO_2$ reduction because of their advantages in wide spectral response and exciton generation[16,17]. By engineering their surface chemistry (e.g., surface defects and/or surface ligands) or introducing proper cocatalysts rationally, photocatalysts based on QDs would markedly boost their catalytic activities[18–20]. Moreover, compared to the zero-

[1]Key Laboratory of Colloid and Interface Chemistry, Ministry of Education, School of Chemistry and Chemical Engineering, Shandong University, 250100 Jinan, P. R. China. ✉e-mail: weijingjing@sdu.edu.cn

dimensional (0D) QDs, one-dimensional (1D) dot-in-rod heterostructures showed a remarkable ability of long-distance separation of photogenerated electrons and holes along their axial direction[21,22]. After proper surface ligands engineering, these QDs-based photocatalysts were able to disperse in a liquid medium under repulsive electrostatic interactions and implement the $CO_2$ photoreduction in solid-liquid mode[23–25]. While in the solid-gas mode, these QDs-based photocatalysts usually suffered from the low $CO_2$ adsorption capacity, which is originated from the fact that packing of colloidal QDs into solids frequently adopts dense, close-packed structures without any open pores[26–28]. In other words, $CO_2$ gases barely interact with the surface layer of the solid catalyst, which thereby markedly reduces the catalytic efficiency. Hence, imparting porosity, especially for hierarchical pores ranging from micropores to mesopores, into the QDs solids would overcome the gas diffusion-limited issue. Packing of QDs into non-close-packed (NCP) structure in confined medium could be one of the possible solutions, but it requires precise design of colloidal building blocks[29,30]. The development of reticular chemistry enables the synthesis of metal-organic frameworks (MOFs), covalent-organic frameworks (COFs), and porous organic cages (POCs), which potentially provides diversified yet robust porous scaffolds hosting the QDs solids[31–35]. Deng and coworkers showed that precise positioning of $TiO_2$ within the MOFs could markedly facilitate $CO_2$ photoreduction benefiting from the synergy between the photoelectron generating $TiO_2$ and the catalytic metals clusters of the MOFs[36]. In comparison with MOFs, COFs, polymerized covalently, showing a superior thermal and chemical stability, could be an intriguing porous host for QDs[37–39]. In this context, the recent development of colloidal COFs provide an ideal platform for hosting the QDs[40–44]. Additionally, the photoactive units and the interlayer stacked noncovalently render COFs semiconducting, which defines pathways for photogenerated charge-carrier transport[45–47]. However, it is still an ongoing challenge to incorporate inorganic nanoparticles (NPs) into COFs with precise positioning of nanoparticles within the COFs[48]. Recently, we have demonstrated that self-assembly of metallic Au NPs is able to synchronize to the polycondensation of COFs, which consequently leads to spatial modulation of Au NPs within the porous matrix[49,50].

Here, we show how the self-assembly of CdSe/CdS dot-in-rod NRs can be in close association with the polycondensation of COFs, which results in self-assembled inorganic/organic double-shelled nanocomposites equipped with micro-meso hierarchical pores. The inner shell is comprised of self-assembled NRs that is in close contact with the outer shell of the COFs, which potentially facilitates the photogenerated electron transfer from NRs to the backbones of the COFs. Moreover, the length of the CdSe/CdS NRs was engineered, which is believed to be correlated with their ability of long-distance separation of photogenerated electrons and holes along their axial direction. Owing to their hierarchical pores and the ability to separate photoexcited electrons, the self-assembled porous nanocomposites exhibit remarkably higher activity ($\approx$64.6 $\mu$mol $g^{-1}$ $h^{-1}$) toward $CO_2$ to CO in solid-gas regime than that of nonporous solids from self-assembled CdSe/CdS NRs or the COFs counterpart under identical conditions.

## Results

### Morphological and structural characterization
We first synthesized the CdSe/CdS NRs coated with octadecylphosphonic acid (ODPA) and hexylphosphonic acid (HPA) ligands via a seeded growth approach, where the length of CdS could be regulated through the stoichiometry control over the seed. As a consequence, a CdSe seed of 3.0 nm in diameter was armored with a shell of CdS of diversified lengths ranging from 12 to 66 nm, denoted by NR12, NR24, NR40 and NR66, as determined from the transmission electron microscopy (TEM) studies (Supplementary Fig. 1). Although their lengths are diversified, the optical properties of these NRs are very similar in terms of both absorption and emission (Supplementary

Fig. 1g, h), which are usually determined by the size of CdSe core. We note that a mixture of phosphonic acid differing by the alkyl chain lengths is crucial to produce NRs with large aspect ratios.

To incorporate these NRs into organic porous matrix to build hierarchically porous nanocomposites, we adopted an emulsion-confined self-assembly/polycondensation process followed by a crystallization process under acidic condition[49,50]. The amine-aldehyde polycondensation reaction catalyzed by a Lewis acid, scandium triflate $(Sc(OTf)_3)$, was principally applied, where C3 symmetric amine-bearing molecules, tris(4-aminophenyl)amine (TAPA), 1,3,5-tris(4-aminophenyl) benzene (TAPB), and 1,3,5-triazine-2,4,6-triyl)trianiline (TAPT), and C2 symmetric aldehyde-bearing 2,5-dimethoxyterephthalaldehyde (DMTA) molecule were used as monomers (Fig. 1a)[51,52]. We took the polycondensation between TAPT and DMTA as a model system to illustrate the fabrication process. Specifically, 2 mL of tetrahydrofuran (THF) solution containing 7 mg of TAPT and 5.8 mg of DMTA was mixed with 6 mL of aqueous solution containing of cationic surfactant solution (dodecyltrimethylammonium bromide, DTAB, 20 g $L^{-1}$) and $Sc(OTf)_3$ (0.5 g $L^{-1}$) to yield oil-in-water (O/W) emulsions after a vortex mixing for 3 min. These micrometer-sized oil droplets dispersed in water serve as microreactors for the polycondensation reactions, which proceed rapidly catalyzed by the Lewis acid and lead to colloidal polymers (TAPT-DMTA) of ~230 nm in diameter within 1 min (Supplementary Fig. 2). Crystallization of these TAPT-DMTA colloids proceeded in a 2 mL mixed solvent of mesitylene/1,4-dioxane (1:8, v/v) at 70 °C for 72 h in the presence of 1 mL of acetic acid aqueous solution of two distinct concentrations (60 vol% or 40 vol%)[44,45]. Remarkably, a low concentration of acetic acid leads to the solid TAPT-DMTA colloids (Supplementary Fig. 3a,b), whereas a higher concentration of acetic acid results in the hollow TAPT-DMTA colloids with a shell thickness of 40 nm (Supplementary Fig. 4), and they were denoted by TAPT-DMTA-S and TAPT-DMTA-H, respectively. The crystallinity of TAPT-DMTA-H is confirmed by the X-ray diffraction (XRD) pattern, where peaks associated with the (100) and (200) facets of TAPT-DMTA can be observed (Supplementary Fig. 4)[53]. The cavitation of TAPT-DMTA colloids under a higher concentration of acetic acid could be driven by the ripening process during the crystallization of polymers, which is frequently observed in the cavitation of inorganic NPs because of the inhomogeneous size of the crystalline domains within a single particle[54,55]. After crystallization, the $N_2$ sorption measurements of both solid and hollow TAPT-DMTA colloids show a classical type I isotherm behavior and specific surface areas calculated from the Brunauer-Emmett-Teller equation ($S_{BET}$) is 622.30 and 470.70 $m^2$ $g^{-1}$[56], respectively, in association with their microporous characteristic (Fig. 2). When the amine-bearing TAPT molecule was replaced with TAPA or TAPB, colloidal polymers of TAPA-DMTA and TAPB-DMTA with similar morphologies could be observed after crystallization at a 60 vol% of acetic acid (Supplementary Fig. 5).

Incorporation of CdSe/CdS NRs into TAPT-DMTA was carried out under identical conditions as described above except that 2 mg of NRs was added into the THF solution additionally. We took NR40 as a typical example to illustrate the structural features of the nanocomposites. The Fourier Transform Infrared (FTIR) spectra revealed that the incorporation of NR40 did not affect the polycondensation reaction apparently (Supplementary Fig. 6). Before crystallization in an acidic solution, nearly all the NR40 were incorporated into the TAPT-DMTA polymers with the formation of colloidal nanocomposites of ~230 nm in diameter. Surprisingly, these NR40 were assembled into a shell-like structure, as evidenced by the sharp contrast in the TEM image (Supplementary Figs. 7a and 8c). Such shell-like structures self-assembled from NR40 could be well reserved after a crystallization process under a 60 vol% of acetic acid, which thereby leads to the TAPT-DMTA/NR40 double-shelled superstructures, denoted by TAPT-DMTA/NR40-H (Fig. 3a, b and Supplementary Figs. 7b and 9f). The crystalline phase of the polymer in TAPT-DMTA/NR40-H is very similar

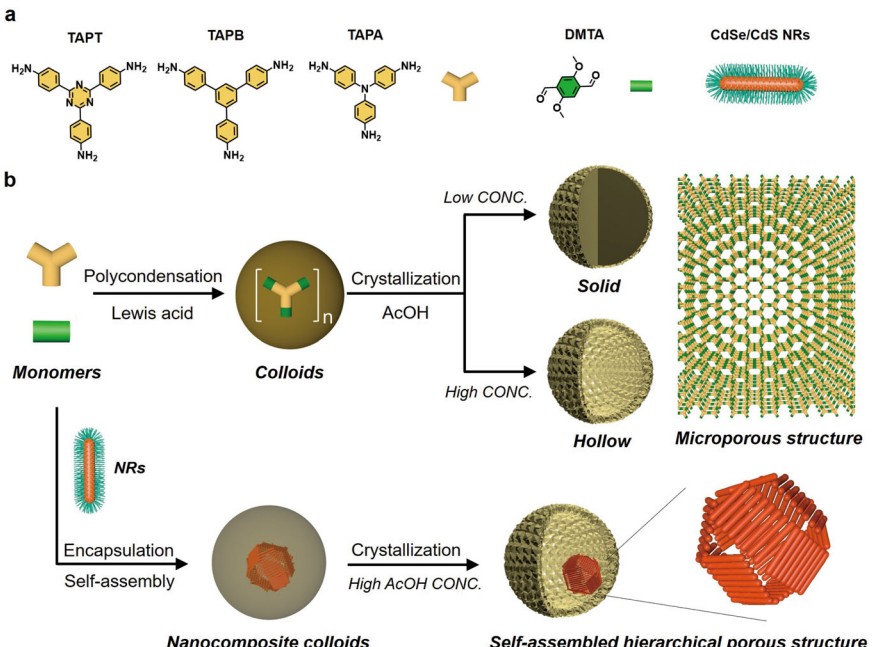

**Fig. 1 | Schematic diagram of the formation of superstructures. a** Schematic diagram of the building blocks used for the synthesis of porous polymers. **b** Design concept for building hierarchically porous nanocomposites from inorganic NRs and porous polymers.

to that of the TAPT-DMTA-H sample, and the crystalline structure of CdS is well reserved, as confirmed by XRD measurements (Supplementary Fig. 10). The inner and outer shell thicknesses in TAPT-DMTA/NR40-H were determined to be $14.6 \pm 3.4$ and $29.5 \pm 3.5$ nm, respectively, by counting more than 200 particles in the TEM images. The detailed microstructure of TAPT-DMTA/NR40-H was further characterized by scanning-TEM energy dispersive X-ray spectroscopy (STEM-EDS) chemical mapping, tilted TEM experiments and focused ion beam scanning electron microscopy (FIB-SEM) in Fig. 3c–e. FIB-SEM image confirmed that nearly all the particles possess a double-shelled hollow structure. STEM-EDS chemical mapping of a single nanocomposite structure revealed that the sulfur element from NR40 is mainly located at the central part, whereas the carbon element from the polymer is mainly distributed at the outer shell of the particle. The tilted TEM experiments further revealed that the self-assembled NR40 superstructures are in close contact with the outer shell of TAPT-DMTA. Additionally, we have conducted high-resolution electron microscopy to understand the interactions between COFs and CdSe/CdS NRs, and the results showed that no apparent gap between TAPT-DMTA COFs and CdSe/CdS NRs could be observed, confirming their intimate contact between the inner shell and the outer shell, which potentially giving rise to the formation semiconducting heterojunction (*vide infra*) (Supplementary Fig. 11). The contact between the NR40 and TAPT-DMTA COFs could be from the atomic interaction between the peripherical amine group (–NH$_2$) of TAPT-DMTA COFs and the surface Cd atoms of CdSe/CdS NRs, which could be verified by the additional experiments described in Supplementary Note 1 and Supplementary Figs. 12 and 13.

The nitrogen sorption isotherm of TAPT-DMTA/NR40-H is identified as a combination mode of type I and type IV with a $S_{BET}$ is 559.62 m$^2$ g$^{-1}$, which is markedly larger than the TAPT-DMTA/NR40 counterpart before crystallization ($S_{BET} = 44.5$ m$^2$ g$^{-1}$, Supplementary Fig. 14). The pore size distribution based on the non-localized density functional theory (NLDFT) clearly shows the mesopores with multiple peaks centered at 7.8, 10.6 and 13.9 nm, respectively (Fig. 2b). We note that the crystallization of TAPT-DMTA/NR40 under a low concentration of acetic acid would lead to the solid colloids (TAPT-DMTA/NR40-S) without the formation of double-shelled superstructures (Supplementary Fig. 3c, d),

and the $S_{BET}$ is markedly reduced to 240.47 m$^2$ g$^{-1}$ (Fig. 2c). The CO$_2$ adsorption isotherms collected at 298 K show distinct affinity between these nanocomposites and CO$_2$ for TAPT-DMTA-S, TAPT-DMTA-H, TAPT-DMTA/NR40-S, and TAPT-DMTA/NR40-H (Supplementary Table 1). The highest CO$_2$ uptake of 13.8 cm$^3$ g$^{-1}$ was observed for TAPT-DMTA/NR40-H, indicating that the as-formed meso-micro hierarchical porous structure promotes the adsorption of CO$_2$. Noteworthy, the encapsulation of self-assembled NR40 hollow superstructures led to a higher CO$_2$ uptake in TAPT-DMTA/NR40-H than that of TAPT-DMTA-H counterpart, despite the fact that the latter has a higher $S_{BET}$ in the nitrogen sorption experiment. Moreover, because of its double-shelled hollow structure, TAPT-DMTA/NR40-H possessed a nontrivial enhanced absorption of visible light than that of TAPT-DMTA-H and TAPT-DMTA/NR40-S counterparts, validated from the UV–vis diffuse reflectance measurements (Fig. 2f, g and Supplementary Fig. 15). Such enhanced absorption in double-shelled hollow superstructures could be due to the strong scattering or diffusing effect when photons transmit through the sample, and the absorption coefficient $\alpha$ could be described as follows:

$$\alpha = -\frac{1}{d}\ln\frac{T}{(1-R)^2}, \tag{1}$$

where $d$, $T$, and $R$ are the shell thickness, transmitted, and reflected incident light, respectively[57]. One could observe that the absorption coefficient $\alpha$ is inversely proportional to the shell thickness of the hollow sphere. The shell thickness of the TAPT-DMTA/NR40-H is merely a few tens of nanometers, which is considered to be responsible for the enhanced absorption of the nanocomposites, akin to miniatured integrating spheres.

To further gain insight into the hollow superstructures assembled from NR40 in the present system, we made several additional observations on the self-assembly of NR40, including (i) NR40 would self-assemble into close-packed structures without the introduction of organic monomers (TAPT or DMTA) (Supplementary Fig. 16). (ii) NR40 were able to assemble into hollow superstructures, which were coated with a thin shell of TAPT-DMTA in the absence of Lewis acid catalyst (Supplementary Fig. 17a, b). (iii) Synthesis/self-assembly in the absence

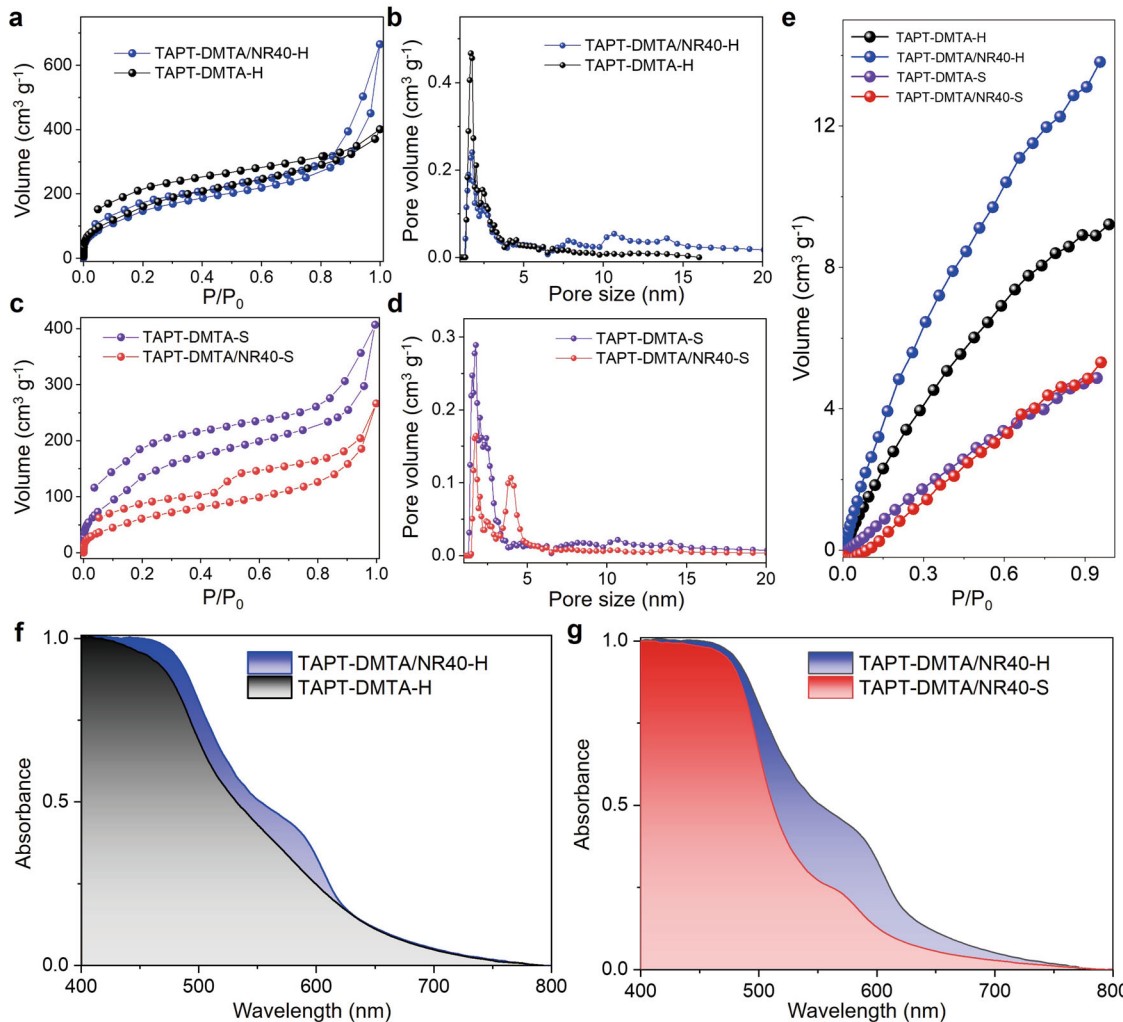

**Fig. 2 | Gas adsorption and light absorption properties.** The $N_2$ adsorption-desorption isotherms (**a**) and pore size distributions (**b**) of TAPT-DMTA-H and TAPT-DMTA/NR40-H. The $N_2$ adsorption-desorption isotherms (**c**) and pore size distributions (**d**) of TAPT-DMTA-S and TAPT-DMTA/NR40-S. The $CO_2$ adsorption curves (**e**) of different samples at 298 K. The normalized solid-state UV–vis absorption spectra (**f**) of TAPT-DMTA-H and TAPT-DMTA/NR40-H. The normalized solid-state UV–vis absorption spectra (**g**) of TAPT-DMTA/NR40-S and TAPT-DMTA/NR40-H.

of the cationic surfactants (DTAB) could not produce colloidal particles with defined morphology, revealing the important role of emulsion droplets in compartmentalization during the synthesis/self-assembly (Supplementary Fig. 17c). (iv) Increasing the temperature for TAPT-DMTA polycondensation, which consequently leads to a higher polycondensation rate, would also result in the self-assembled NR40 hollow superstructures with a thicker TAPT-DMTA polymer coating (Supplementary Fig. 18). (v) The polycondensation of TAPT-DMTA and self-assembly of NR40 polymer was synchronized and the hybridized colloids were rapidly formed without the removal of THF under evaporation process (Supplementary Fig. 19). (vi) when the TAPT monomer was replaced with TAPA or TAPB monomer, similar double-shelled superstructures could be produced under the identical conditions (Supplementary Fig. 20). The above observations allow us to conclude that self-assembled NR hollow superstructures could be templated by the liquid-liquid interface and subsequently be solidified by the interfacial polycondensation reactions. Likewise, when NR40 was replaced with other NRs differing by their lengths, TAPT-DMTA/NR12-H, TAPT-DMTA/NR24-H with similar double-shelled superstructures could be produced after crystallization at 60 vol% of acetic acid (Supplementary Fig. 9). The $CO_2$ uptake experiments showed that the $CO_2$ uptake for TAPT-DMTA/NR12-H and TAPT-DMTA/NR24-H is 12.4 and 10.4 cm³ g⁻¹

(Supplementary Fig. 21), respectively, which is very close to that of the TAPT/DMTA/NR40-H (13.8 cm³ g⁻¹). While for TAPT-DMTA/NR66-H sample, most of the NR66 prefer to self-assemble in a side-by-side mode, which hampers the formation of hollow superstructures.

## Charge transfer process
To get insight into the photo-induced charge separation efficiency of self-assembled double-shelled hollow superstructures, steady-state and time-resolved photoluminescence (PL), photocurrent measurements and electron paramagnetic resonance (EPR) were performed in Fig. 4a–f. First, we prepared NR40 assemblies in the absence of TAPT-DMTA, which showed strong PL intensity in the steady-state PL spectrum (Fig. 4a). In sharp contrast, the TAPT-DMTA/NR40-H sample in the presence of TAPT-DMTA showed marked decrease in PL intensity, as shown in Fig. 4a, suggesting that a photo-induced charge transfer takes place. To further clarify the charge transfer process, we conducted the exciton lifetime measurement. In the absence TAPT-DMTA, the CdSe/CdS NR40 assemblies exhibited triple exponent decay with time constants of $\tau_1$ = 9.52 ns (49.22%), $\tau_2$ = 29.86 ns (37.03%) and $\tau_3$ = 176.14 ns (13.75%) (Fig. 4b). The calculated average time of $\tau_{ave}$ was about 39.96 ns. However, the emission decay becomes faster in the presence of TAPT-DMTA. In the TAPT-DMTA/NR40-H sample, the CdSe/CdS NR40 exhibited a fast triple exponent

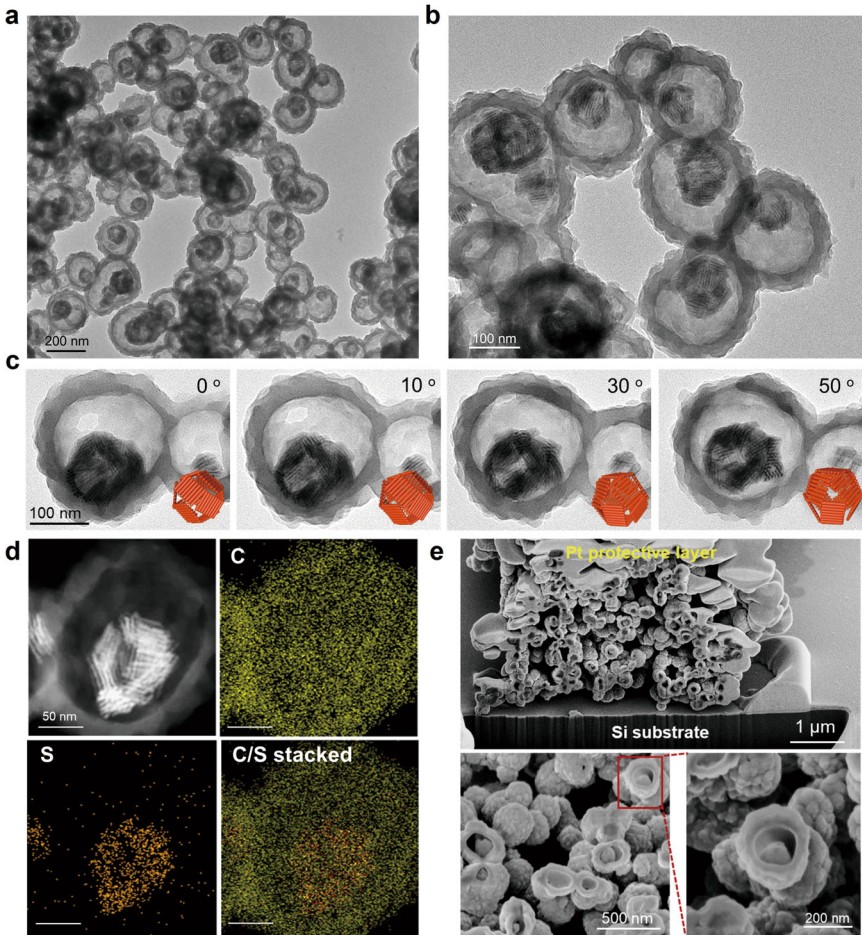

**Fig. 3 | Morphological characterization of the double-shelled nanocomposites. a, b** TEM images of TAPT-DMTA/NR40-H. **c** TEM images of TAPT-DMTA/NR40-H recorded at different tilting angles from 0º to 50º. **d** The HAADF-STEM image and EDS elemental maps of TAPT-DMTA/NR40-H. **e** The FIB-SEM images of TAPT-DMTA/NR40-H.

decay with time constants of $\tau_1 = 0.03$ ns (57.84%), $\tau_2 = 0.28$ ns (29.40%), $\tau_3 = 0.96$ ns (12.77%) and the calculated average time of $\tau_{ave}$ was about 0.22 ns (Fig. 4c). Additionally, compared to that of the TAPT-DMTA/NR40-S solid counterpart ($\tau_{ave} = 0.55$ ns), the exciton lifetime of the TAPT-DMTA/NR40-H was shorter. Moreover, when the length of CdSe/CdS NRs was reduced, the calculated average time of $\tau_{ave}$ for TAPT-DMTA/NR12-H and TAPT-DMTA/NR24-H is 0.58 and 0.25 ns, respectively, which is slightly longer than that of the TAPT-DMTA/NR40-H, suggesting that the photo-induced charge separation in the nanocomposites with shorter NRs is less efficient (Supplementary Fig. 22 and Supplementary Table 2). To further unveil the impact of the NR length on the photo-induced charge separation efficiency, we measured the photocurrents under visible light irradiation and the ESR spectra for various double-shelled superstructures differing by the NR length. The photocurrent measurements showed that all the double-shelled superstructures displayed fast response to the incident light with remarkably larger photocurrent compared to that of TAPT-DMTA-H counterpart in the absence of NRs (Fig. 4d). More interestingly, the increase of the length of NRs from 12 to 40 nm would increase the intensity of the photocurrents, following an increasing order of TAPT-DMTA/NR12-H ($0.68 \pm 0.02$ μA cm$^{-2}$), TAPT-DMTA/NR24-H ($1.56 \pm 0.03$ μA cm$^{-2}$) and TAPT-DMTA/NR40-H ($2.15 \pm 0.10$ μA cm$^{-2}$) (Fig. 4e). Additionally, such a trend in photo-induced charge separation efficiency was further confirmed by the EPR data, where the strongest signal was produced from TAPT-DMTA/NR40-H after light irradiation for 5 min (Fig. 4f).

Next, we set out to identify the band position of nanocomposite TAPT-DMTA/NR40-H, which is comprised of three semiconducting components: CdSe, CdS, and TAPT-DMTA. As is well known, CdSe/CdS dot-in-rod structure is a typical Type-I heterojunction, where the conduction band (CB) and valence band (VB) of CdS straddle the narrower band gap of CdSe[21]. The CdS domain in the CdSe/CdS heterostructures can intensely absorb light and transport the light energy into the CdSe domain, where photochemical reactions or photoluminescence can occur. Herein, we introduced another organic semiconductor TAPT-DMTA, which dramatically impacts the pathway of excitons/charge carriers. We measured the flat band potentials ($V_{fb}$) of the semiconductors by the Mott-Schottky plot from the electrochemical approach. All the CdSe, CdS NRs, and TAPT-DMTA exhibited positive slopes in the Mott-Schottky plots, a characteristic typical of an n-type semiconductor[58]. We applied three different angular frequencies (1000, 1500, and 2000 Hz) of the voltage to determine the $V_{fb}$ of semiconductors, and the values were determined to be −0.74, −0.97, and −0.65 V vs. NHE for CdSe, CdS, and TAPT-DMTA, respectively (Supplementary Fig. 23). In general, the bottom of conduction band or lowest unoccupied molecular orbital (LUMO) stays very close to the flat band potential in *n*-type semiconductors[58,59]. To a first approximation, we used the flat band potential as their CBs. The HOMO level or VBs of the semiconductors could be calculated by considering their respective optical band gaps. The optical band gaps, determined from the Tauc plots, are 2.21, 2.61, and 2.33 eV for CdSe, CdS, and TAPT-DMTA, respectively (Supplementary Fig. 24). Therefore, the band positions of three

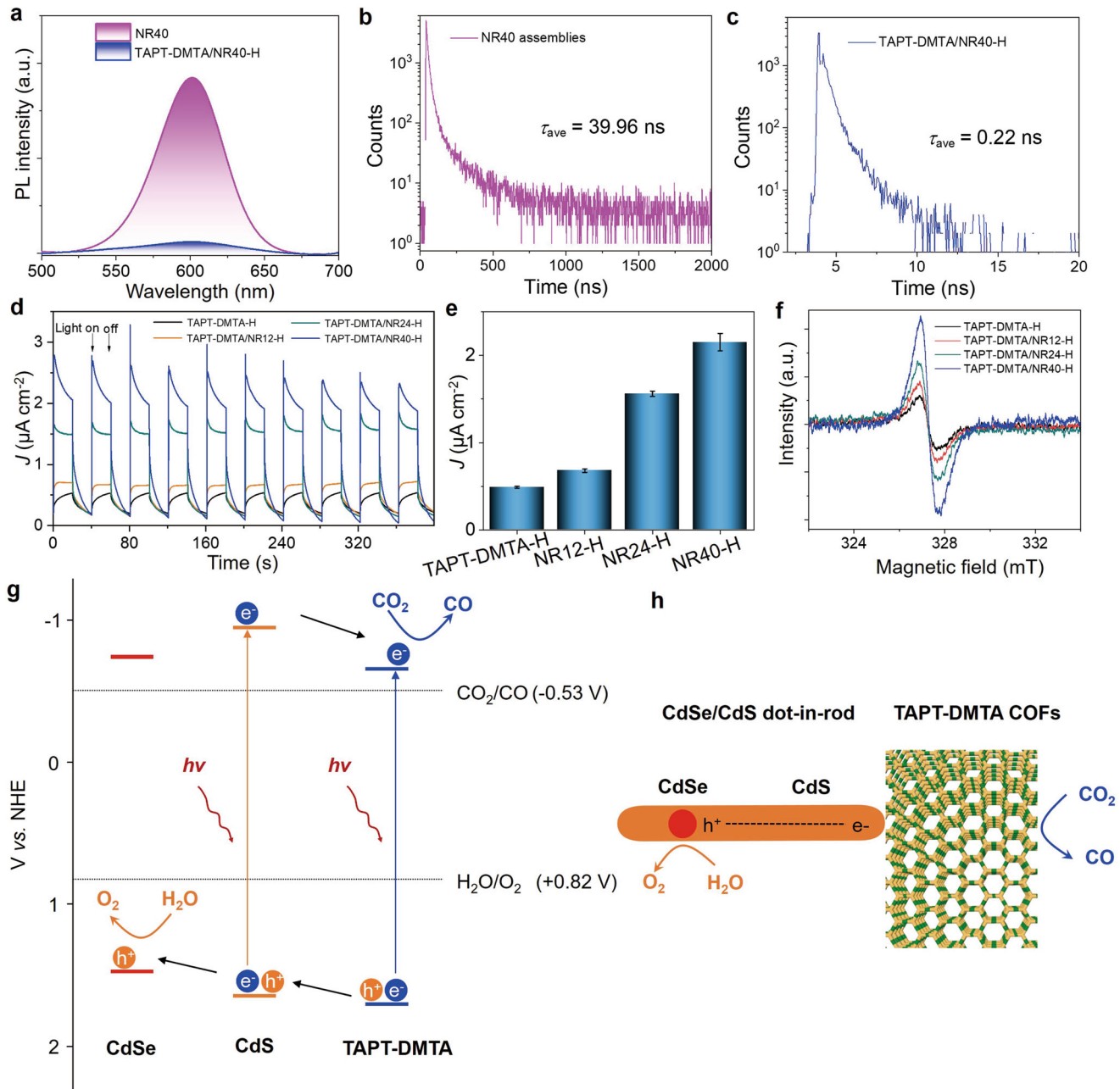

**Fig. 4 | Photogenerated charge transfer properties. a** Steady-state PL spectra of NR40 and TAPT-DMTA/NR40-H. **b** Time-resolved PL spectra of NR40 assemblies. **c** Time resolved PL spectra of TAPT-DMTA/NR40-H. **d** Transient photocurrent response and the corresponding data statistics (**e**) of TAPT-DMTA-H, TAPT-DMTA/NR12-H, TAPT-DMTA/NR24-H, and TAPT-DMTA/NR40-H; error bars are standard deviations calculated from three independent measurements. **f** EPR spectra of TAPT-DMTA-H, TAPT-DMTA/NR12-H, TAPT-DMTA/NR24-H, and TAPT-DMTA/NR40-H. **g** Band-structure diagram of TAPT-DMTA, CdS and CdSe. **h** Proposed $CO_2$ photoreduction reaction in the presence of heterojunctions in the TAPT-DMTA/NR40-H nanocomposites.

semiconductors in the nanocomposites could be determined (Fig. 4g). Concerning on the fact that CB of CdS is more negative than that of TAPT-DMTA, whereas VB of TAPT-DMTA is more positive than that of CdS, forming a typical Type-II heterojunction between CdS NR and TAPT-DMTA. We speculate that the photogenerated charge could be separated efficiently—the electron transfer from CdS to TAPT-DMTA and the hole transfer TAPT-DMTA to CdS and to CdSe core. Hence, a donor (CdSe)-absorber (CdS)-acceptor/catalyst (TAPT-DMTA) inorganic/organic nanocomposite superstructure is constructed and, the distance of charge separation in the charge separated state ($CdSe^+$-CdS-TAPT-DMTA$^-$) can be simply controlled by the CdS rod length, which is experimentally confirmed by the results of PL lifetime, photocurrents and EPR data.

Given that the CB of the TAPT-DMTA is more negative than the theoretical $CO_2$-to-CO reduction potential, $CO_2$-to-CO photoreduction is possible on the one hand. On the other hand, the VB of the CdSe or CdS is more positive than the theoretical water oxidation reaction (oxidation of $H_2O$ to $O_2$). Therefore, these double-shelled super-structures have great potential to complete the overall $CO_2$ photo-reduction in the solid-gas regime by using $H_2O$ as the electron donor (Fig. 4h). Based on the energy level of the CdSe, CdS, and TAPT-DMTA COFs, we believe that the $CO_2$ photoreduction takes place at the TAPT-DMTA. To further identify the active site of the TAPT-DMTA COFs, we performed the density functional theory (DFT) calculations on the unit of the TAPT-DMTA COFs, which gives the information of frontier molecular orbitals, highest occupied molecular orbital (HOMO)-LUMO

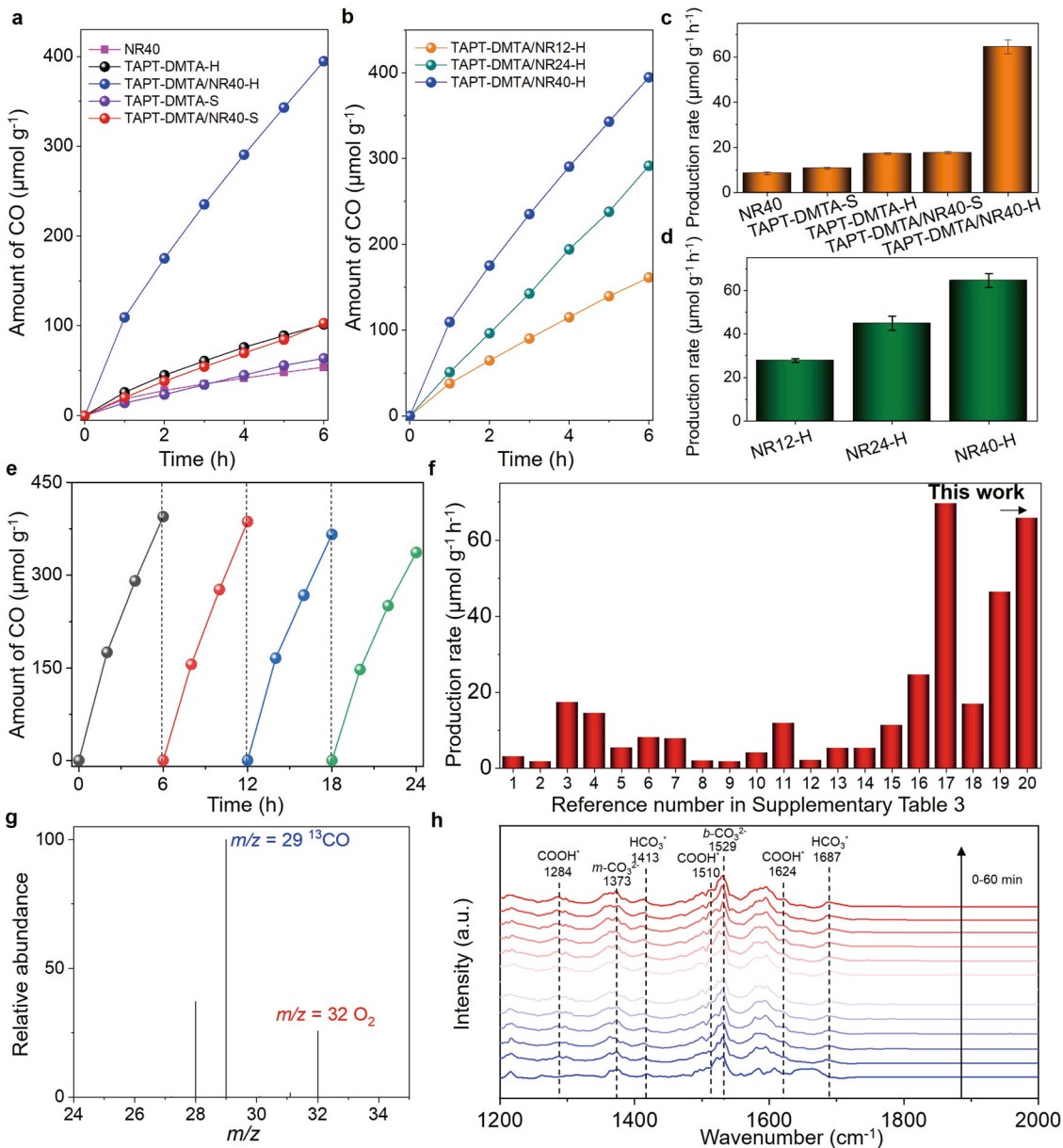

**Fig. 5 | CO$_2$ photoreduction performances of the catalysts. a** Time-dependent CO$_2$-to-CO performances of NR40, TAPT-DMTA-H, TAPT-DMTA/NR40-H, TAPT-DMTA-S, and TAPT-DMTA/NR40-S. **b** Time-dependent CO$_2$-to-CO performances of TAPT-DMTA/NR12-H, TAPT-DMTA/NR24-H, and TAPT-DMTA/NR40-H. **c, d** The CO$_2$-to-CO generation rate of different samples, the error bars were determined by three independent catalytic runs. **e** CO$_2$ photoreduction cycle performance of TAPT-DMTA/NR40-H. **f** Comparison of the efficiency of recently published references on photoreduction CO$_2$ in gas-solid systems. Detailed information is shown in Supplementary Table 3. **g** GC-MS analysis of $^{13}$CO$_2$ isotope after $^{13}$CO$_2$ photoreduction in the presence of TAPT-DMTA/NR40-H. **h** In-situ DRIFTS of TAPT-DMTA/NR40-H in the presence of CO$_2$ and H$_2$O vapor within 60 min illumination.

energy level and the molecular electrostatic potentials. We found that the more negative electrostatic potential could be found in the imine and triazine positions, suggesting that these positions are most probably catalytically active sites for the CO$_2$ reduction reaction (Supplementary Fig. 25).

## Photocatalytic performance and reaction mechanism
Encouraged by the CO$_2$ uptake capacity, strong visible light absorption, and efficient photoexcited charge separation in the self-assembled doubled-shelled nanocomposites, we investigated the CO$_2$ photoreduction in the gas-solid regime under visible light irradiation ($\lambda > 420$ nm), without additional photosensitizer or sacrificial agent. The detailed descriptions of the measurements of the CO$_2$ photoreduction are shown in Supplementary Information. All the

solid-state photocatalysts were activated by soaking treatments with CH$_3$OH, followed by the baking treatment at 70 °C for 24 h. The online gas-chromatographic (GC) analysis of the product was exemplified in Supplementary Fig. 26. In the first set of experiments, we showed that TAPT-DMTA/NR40-H displays an optimal CO yield of 395 μmol per gram of nanocomposite after 6 h (simplified as μmol g$^{-1}$) at a given mass ratio (TAPT-DMTA:NR40 = 12.8:2) in Supplementary Fig. 27. Importantly, the TAPT-DMTA/NR40-H sample shows a markedly higher CO yield than 53.9 μmol g$^{-1}$ for NR40, 63.8 μmol g$^{-1}$ for TAPT-DMTA-S, 101.6 μmol g$^{-1}$ for TAPT-DMTA-H, and 103 μmol g$^{-1}$ for TAPT-DMTA/NR40-S samples after 6 h (Fig. 5a). Particularly, the time-dependent CO$_2$-to-CO performances of TAPT-DMTA/NR40-H, TAPT-DMTA/NR24-H and TAPT-DMTA/NR12-H were conducted. Based on the three independent runs, we evaluated the data statistics, which

showed that the error bar of the CO formation rate for each sample is less than ±10 %, revealing the good reproducibility of the photocatalysts (Supplementary Fig. 28). This trend can be further interpreted by the average CO formation rate. The average CO formation rate, determined by three catalytic runs, for NR40 assemblies, TAPT-DMTA-S, TAPT-DMTA-H, TAPT-DMTA/NR40-S, TAPT-DMTA/NR40-H, are 8.5, 10.7, 17.2, 17.6, and 64.6 µmol g$^{-1}$ h$^{-1}$, respectively (Fig. 5c). The high CO formation rate in TAPT-DMTA/NR40-H sample is due to the double-shelled hollow superstructures with both high $CO_2$ uptake and strong visible light absorption. We note that the CO formation rate of the TAPT-DMTA/NR40-H sample (64.6 µmol g$^{-1}$ h$^{-1}$) without any additional photosensitizer or sacrificial agent is one of the best performances reported for the $CO_2$ photoreduction in the solid-gas regime (Fig. 5f and Supplementary Table 3). It is important to note that no apparent $H_2$ gas could be detected, revealing the excellent selectivity of the TAPT-DMTA/NR40-H for the $CO_2$ photoreduction. In another set of experiments, we studied how the length of NR impacts on the CO formation rate during the $CO_2$ photoreduction under the identical conditions. Figure 5b shows that the length of the NR could strongly impact on the CO yield after 6 h, and a longer NR in the nanocomposites leads to a higher yield, which consequently results in a faster CO formation rate (Fig. 5d). This observation is in line with the EPR analysis and time-resolved PL data, which confirmed that a longer CdSe/CdS NR in the nanocomposites could lead to more efficient photogenerated charge separation. Equally important, the activity of TAPT-DMTA/NR40-H toward $CO_2$ photoreduction maintained even after four catalytic cycles, and their morphology did not show any detectable change after catalytic cycles, confirming their excellent durability during the $CO_2$ photoreduction in the solid-gas regime (Fig. 5e and Supplementary Fig. 29). We note that a slight decrease in the activity after the third cycle could be observed, which is probably due to the formation of surface products and the lack of any catalyst regeneration process. External quantum efficiency (EQE) tests were conducted on TAPT-DMTA/NR40-H (Supplementary Fig. 30 and Supplementary Table 4) and the EQE trend matched with photoabsorption spectrum of the heterogeneous potocatalyst[15]. We have performed the $CO_2$ photoreduction without any sacrificial agents. To confirm another half-reaction, we used 50 mg of the TAPT-DMTA/NR40-H photocatalyst to conduct the $CO_2$ photoreduction, and the products were detected by GC. The standard curve for oxygen gas was first depicted by GC, which was applied to quantify the amount of oxygen generated during the water oxidation reaction. The result in Supplementary Fig. 31 revealed that the stoichiometry between CO and $O_2$ gas is measured to be around 2:1 at three different reaction times. Hence, we conclude that another half-reaction is the oxygen evolution from water oxidation reaction.

To identify the origin of the product of CO, we performed the isotope experiment by using $^{13}CO_2$ as reactant to proceed the photoreduction reaction under the identical conditions. The as-formed gas product was validated by gas chromatography mass spectroscopy (GC-MS). The MS spectra revealed that $^{13}CO$ is the main product, which is in close association with the $^{13}CO_2$ photoreduction in Fig. 5g. Additionally, the peak of $m/z = 32$ could be associated with the oxygen from the oxidation of water during the photocatalysis[60–62]. Furthermore, we employed in-situ diffuse reflectance infrared Fourier transform spectroscopy (DRIFTS) to track the reaction intermediates during the adsorption, activation, and reduction of $CO_2$ on the surface of TAPT-DMTA/NR40-H under visible light irradiation (Fig. 5h). The observation of new infrared peaks at 1284, 1510, and 1624 cm$^{-1}$ was attributed to the COOH* (the asterisk denotes the catalytically active sites) group, which is generally regarded as the important intermediate during $CO_2$ reduction to CO[63–65]. The peaks at 1413 and 1687 cm$^{-1}$ were attributed to the symmetric stretching and asymmetric stretching of HCO$_3$*, respectively[63]. The formation of monodentate carbonate (m-CO$_3^{2-}$) and bidentate carbonate (b-CO$_3^{2-}$) was evidenced from the infrared peaks

of 1373 and 1529 cm$^{-1}$, respectively. These data suggest that the formation of COOH* intermediates from the TAPT-DMTA could be plausible for the production of CO during the $CO_2$ photoreduction. Noteworthy, the triazine group from the TAPB-DMTA could be of particular importance for the reduction reactions. Replacing the TAPT monomer with TAPA or TAPB, while keeping the other conditions constant, results in the nanocomposites of TAPA-DMTA/NR40-H or TAPB-DMTA/NR40-H, respectively. The $CO_2$ photoreduction experiments revealed that the CO formation rate for TAPA-DMTA/NR40-H and TAPB-DMTA/NR40-H is 10 and 15.8 µmol g$^{-1}$ h$^{-1}$, which is much slower than that of the TAPT-DMTA/NR40-H nanocomposites (Supplementary Fig. 32). Moreover, the poor catalytic performances of TAPT-DMTA/NR40 before crystallization in Supplementary Fig. 33 indicate that the efficient crystallization of polymer is critical.

## Discussion

In summary, we have presented a rational strategy for efficient $CO_2$ capture and photoreduction by hierarchically porous materials comprised of both inorganic and organic semiconductors. Synchronizing imine polycondensation of organic polymers to self-assembly of inorganic CdSe/CdS NRs at the liquid-liquid interface is crucial to achieve the double-shelled hollow superstructures. In combination of several remarkable features within the superstructures including (1) enhanced visible light absorption from the double-shelled structures, (2) high $CO_2$ uptake capacity from the micro-meso hierarchically porous structures, and (3) efficient photogenerated charge separation from the large aspect ratio of inorganic NRs, they display a high activity toward $CO_2$ photoreduction to CO (64.6 µmol g$^{-1}$ h$^{-1}$). Looking forward, beyond the CdSe/CdS NRs, other colloidal inorganic NPs, such as metallic Au, Pt, Pd and semiconducting CdS, were able to be encapsulated to the microporous polymers under the present synthetic approach, leading to the double-shelled inorganic/organic superstructures (Supplementary Fig. 34). We envisage that a productive avenue for future research is the use of double-shelled inorganic/organic superstructures to direct the chemical reactions in a sequential fashion.

## Methods

### Materials

Cadmium oxide (CdO), octadecylphosphonic acid (ODPA) and hexylphosphonic acid (HPA) were purchased from Sigma-Aldrich. Sulfur powder (S), selenium powder (Se), trioctylphosphine (TOP), trioctylphosphine oxide (TOPO) were obtained from Macklin. Scandium triflate (Sc(OTf)$_3$) was purchased form TCI. Tris(4-aminophenyl)amine (TAPA), 1,3,5-tris(4-aminophenyl) benzene (TAPB), and 1,3,5-triazine-2,4,6-triyl)trianiline (TAPT) and 2,5 dimethoxyterephthalaldehyde (DMTA) were purchased from Yanshen Technology. Ethanol, methanol, hexane, toluene and tetrahydrofuran (THF) were obtained from Sinophram. All chemicals are used without further purification.

### Synthesis of CdSe seeds

The CdSe seeds are synthesized by the modified reaction[66]. CdO (0.030 g), ODPA (0.150 g) and TOPO (2.0 g) are mixed in a 50 mL four flask, degassing for 10 min in a vacuum at room temperature. The temperature is then slowly increased to 120 °C and kept at this temperature for 30 min. Then, under nitrogen, the solution is heated to above 300 °C to dissolve the CdO until it turns optically clear and colorless. At this point, 1.0 g of TOP is quickly injected into the flask. When the temperature reaches 370 °C, the Se: TOP solution (0.029 g Se + 0.220 g TOP) is injected, and the heating is stopped immediately after injection and allowed to cool naturally. When the reaction solution cooled to room temperature, toluene is added to disperse it. Excess ethanol is added to precipitate the nanocrystals, and the nanocrystals are purified by centrifugation, repeated twice, and the final nanocrystals are dispersed in 5 ml of TOP.

## Synthesis of CdSe/CdS nanorods

The CdSe/CdS nanorods are prepared through a modified approach as reported[66]. In a typical synthesis of CdSe/CdS nanorods, CdO (0.030 g), ODPA (0.150 g), TOPO (1.0 g), and HPA (0.040 g) are mixed into a 50 mL four flask, degassing for 10 min in vacuum at room temperature. The temperature is then slowly increased to 120 °C for 30 min. Then, under nitrogen, the solution is heated to above 300 °C to dissolve the CdO until it turns optically clear and colorless. At this point, 1.0 g of TOP is quickly injected in the flask. When the temperature reaches 350 °C, the mixed solution containing S: TOP (0.060 g S + 1.0 g TOP) and CdSe seeds are quickly injected. When the temperature returns to 350 °C, keep it for 10 min. Remove heat source and cool naturally. When the reaction solution cooled to room temperature, toluene is added to disperse it. The nanocrystals are washed by cycled toluene/ethanol, and they are finally dissolved in 5 mL of toluene. The length of CdSe/CdS nanorods can be regulated through the stoichiometry control over the seed. As a consequence, the added amount of seeds is 900, 600, 300, and 200 µL, respectively, which results in nanorods with a CdSe seed of 3 nm in diameter and a shell of CdS of diversified lengths ranging from 12 to 66 nm, denoted as NR12, NR24, NR40, and NR66, respectively.

## Polycondensation and crystallization of TAPT-DMTA

The TAPT-DMTA colloids are prepared through an O/W emulsion technique[49,50]. Typically, TAPT (7 mg, 0.02 mmol) and DMTA (5.8 mg, 0.03 mmol) are dissolved in 2 mL of THF to form a transparent solution. 6 mL of DTAB aqueous solution (20 g L$^{-1}$) containing Sc(OTf)$_3$ (0.5 g L$^{-1}$) is then quickly added to the above solution. The resulting mixture is severely agitated by a vortex for 3 min and then heated to 70 °C and kept at this temperature for 30 min to evaporate the THF phase. After cooling to room temperature, the suspension is washed twice with water and dried under vacuum. The crystallization of TAPT-DMTA is performed in a glass vial containing a mixture of dioxane/mesitylene (4:1 v/v, 2 mL), H$_2$O (0.4 mL) and CH$_3$COOH (0.6 mL) at 70 °C for 72 h. The crystallized TAPT-DMTA colloids are washed several times via centrifugation (7871 g, 3 min) with methanol, and subsequently dried under vacuum. The high concentration of acetic acid (60 vol%) results in the hollow TAPT-DMTA colloids with a shell thickness of 40 nm and they are denoted as TAPT-DMTA-H.

## Synthesis of TAPT-DMTA/NRs

A similar O/W emulsion technique is used for synthesis of TAPT-DMTA/NRs composites with adding CdSe/CdS nanorods with different lengths in the THF phase. Typically, TAPT (7 mg, 0.02 mmol), DMTA (5.8 mg, 0.03 mmol), and CdSe/CdS nanorods (2 mg) are dissolved in 2 mL THF to form a transparent solution. 6 mL DTAB solution (20 g L$^{-1}$) containing 0.5 g L$^{-1}$ of Sc(OTf)$_3$ is then quickly added to the above solution. The resulting mixture is severely agitated by a vortex for 3 min and then heated to 70 °C and kept at this temperature for 30 min to evaporate the THF. After cooling to room temperature, the suspension is washed twice with water and dried under vacuum. The crystallization of TAPT-DMTA/NRs is applied in a glass vial with a mixed solution containing dioxane/mesitylene (4:1 v/v, 2 mL), H$_2$O (0.4 mL) and CH$_3$COOH (0.6 mL) at 70 °C for 72 h. The TAPT-DMTA/NRs solid is washed several times via centrifugation (7871 g, 3 min) with methanol, and subsequently dried under vacuum. The high concentration of acetic acid (60 vol%) results in the hollow TAPT-DMTA/NRs and they are denoted as TAPT-DMTA/NR12-H, TAPT-DMTA/NR24-H, TAPT-DMTA/NR40-H and TAPT-DMTA/NR66-H based on the length of CdSe/CdS nanorods.

## Synthesis of solid TAPT-DMTA and solid TAPT-DMTA/NR40

The synthetic procedure of solid TAPT-DMTA and solid TAPT-DMTA/NR40 are similar to the TAPT-DMTA-H and TAPT-DMTA/NR40-H except for the crystallization process. Typically, the crystallization of TAPT-DMTA and TAPT-DMTA/NR40 are applied in a bottle with the mixed solution containing dioxane/mesitylene (4:1 v/v, 2 mL), H$_2$O (0.6 mL) and CH$_3$COOH (0.4 mL) at 70 °C for 72 h. The low concentration of acetic acid (40 vol%) results in the solid colloids and they are denoted as TAPT-DMTA-S and TAPT-DMTA/NR40-S.

## Characterization

The QuadraSorb SI is used in order to analyze the specific surface area, pore size, and CO$_2$ adsorption quantity. UV−vis diffuse reflection absorption spectra of samples are obtained by the UV2600 spectrophotometer equipped with an integrating sphere accessory and Teflon as a reference material. The morphologies and microstructures are inspected by transmission electron microscopy (TEM, HT-7700, accelerating voltage 100 kV) and scanning electron microscope (FE-SEM, ZEISS SUPRATM 55). Elemental characterization including HAADF and EDS images are carried out by using STEM (FEI Tecnai G2 TF20). Powder X-ray diffraction (PXRD) is recorded using a Bruker D8 DISCOVER X-ray power diffractometer. Time-resolved fluorescence decay is measured by time-correlated-single photon counting (Edinburgh Instruments, FLS 920). Electron paramagnetic resonance (EPR) signals are recorded using JEOL JES-X320. Photoluminescence (PL) measurements are performed at room temperature using a F-320 fluorescence spectrophotometer.

## Photocatalytic CO$_2$ reduction measurement

In the typical process, 5 mg of catalyst is dispersed in ethanol, and then the dispersions are dropped onto the filter paper with a diameter of 4 cm. After the filter paper is dried at 70 °C, it is put into the reaction vessel with 3 mL of deionized water on the bottom. The volume of the reaction system (Labsolar-6A, Beijing Perfectlight Technology Co., Ltd.) is about 400 mL. The reaction system is treated three times by vacuuming, and the pressure of high-purity CO$_2$ and H$_2$O vapor reaches 70 KPa. A 300 W Xe lamp with 420 nm filter is placed on the top of the reactor as the light source, and the distance between the light source and the filter paper is approximately 10.0 cm. During irradiation, the temperature of the reaction system is controlled at about 8 °C through the circulating cooling water system. During the reaction process, the product gases are qualitatively analyzed by the online gas chromatograph (GC9790II, PLF-01) equipped with a flame ionization detector (FID) and thermal conductivity detector (TCD). In order to determine the source of carbon in the product, $^{13}$CO$_2$ (99% in purity) is used for isotope experiments under identical photocatalytic reaction conditions. GC-MS result of $^{13}$CO$_2$ isotope experiment is detected by GC-MS Spectrometer (GC-MS, Agilent 8860-5977B GCMS).

## Reporting summary

Further information on research design is available in the Nature Research Reporting Summary linked to this article.

## Data availability

The data that support the finding of this study are available from the corresponding author upon request. Source data are provided with this paper.

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

## Acknowledgements

This work was financially supported by the National Natural Science Foundation of China (Grant No. 21972076) and Shandong Provincial Natural Science Foundation (Grant Nos. ZR2020YQ11 and ZR2021JQ05). We thank Prof. P. Li in Shandong University for the gas adsorption measurements.

## Author contributions

H.L. and C.K.C. performed the experiments. H.L., C.K.C., Z.J.Y., and J.J.W. discussed and analyzed the data. J.J.W. conceived and supervised the project. J.J.W. wrote the manuscript and all the authors commented on to the manuscript.

## Competing interests

The authors declare no competing interests.

## Additional information

**Correspondence and requests** for materials should be addressed to Jingjing Wei.

