## [Peer Review File · Nature Communications]

Efficient CO₂ photoreduction with CdS/CdSe nanorods encapsulated in double-shelled porous nanocompositesReviewer #1 (Remarks to the Author):

Dr. Wei and co-workers synthesized COF-based nanorod composites and fully characterized them. One of the COF composites in the series shows excellent photocatalytic CO₂ reduction under the gas phase without any sacrificial agents. I recommend publishing this work after a major revision. Below are my comments and suggestion.

- 1) Lines 89, 90, "CdSe/CdS coated with octadecylphosphonic acid (ODPA) and hexylphosphonic acid (HPA) ligands via a seeded growth approach", can the authors elaborate more on the role of ODPA and HPA?**
- 2) Lines 106, 107, what is the role of surfactant DTAB?**
- 3) Sc(OTf)₃ plays as a Lewis catalyst. Why do the authors have to use AcOH 40% and 60% to catalyze the imine-linked COFs formation?**
- 4) The authors should double-check the inner and outer shell thicknesses in TAPT-DMTA/NR40-H. If based on Fig 9f, I don't think it is 8 and 30 nm, respectively.**
- 5) The authors mentioned high CO₂ uptake in the introduction. However, I don't think the CO₂ uptake of the COF composite materials in this work is high (the best one in the series is just 13.8 cc/g). Therefore, it is not convincing to say, "indicating that the as-formed meso-micro hierarchical porous structure promotes the adsorption of CO₂." (line 161).**
- 6) The authors should elaborate more on how to determine conduction band based on the flat band and bandgap energy.**
- 7) Can the authors explain a bit more on Fig. 4 e and f? How to interpret to get 63.06 and 0.22 ns?**
- 8) It is useful to readers if the authors explain why the longer length of NR will promote the photoexcited charge separation efficiently?**
- 9) Line 225, the authors should consider the CO₂ uptake as discussed above.**
- 10) "Denoted as" is just fine, but I would use "denoted by". Also, lines 125, 183, and 187: it should be "replaced with" instead of "replaced by".**
- 11) I think the catalytic performance of TAPT-DMTA/NR40-H starts dropping from the third cycle. The authors should explain and revise the manuscript accordingly.**
- 12) I suggest the author cite relevant reviews about using COFs for CO₂ photoreduction recently published by Nguyen in ACS Catalyst or Advanced Energy Materials (ACS Catal. 2021, 11, 9809–9824; Adv. Energy Mater. 2020 10, 2002091).**

Reviewer #2 (Remarks to the Author):

In this work, the authors demonstrate the design and fabrication of integrated organic and inorganic semiconductors for CO₂ photoreduction. The authors incorporate self-assembled CdSe/CdS dot-in-rod nanorods within the micro-meso porous COFs matrix, resulting in a double-shelled inorganic/organic nanocomposites. In addition, the authors investigate how the length of the CdSe/CdS nanorods effects the long distance separation of photo-induced electrons and holes. However, the authors didn't clearly validate the interaction between the CdSe/CdS nanorods and COFs matrix with related measurements. The gas-solid system of the CO₂ photoreduction reaction in this manuscript also needs more proof of the complete reactions. Overall, the claims are poorly supported by solid evidence. Therefore, I would not recommend it to be published in this journal. The following are some suggestions for the authors to improve the quality of this paper for more-specialized journals.

1. Self-assembly of the CdSe/CdS dot-in-rod nanorods have been widely studied in previous works(Synthesis and micrometer-scale assembly of colloidal CdSe/CdS nanorods prepared by a seeded growth approach, Nano letters, 2007, 7, 2942; CdSe and CdSe/CdS Nanorod Solids, J. Am. Chem. Soc. 2004, 126, 12984), and the structure of the TAPA/TAPB/TAPT-DMTA COFs used in this work is also examined explicitly (Stable, crystalline, porous, covalent organic frameworks as a platform for chiral organocatalysts. Nature Chemistry, 2015, 7, 905; Using sound to synthesize covalent organic frameworks in water, Nature Synthesis, 2022, 1, 87). But the authors combine

these two semiconductors together without highlighting the interaction between these two components.

2. The authors claim that "The inner shell is comprised of self-assembled NRs that is in close contact with the outer shell of the COFs, which potentially facilitates the photogenerated electron transfer from NRs to the backbones of the COFs." However, the NRs do not atomically interact with COFs, and are only partially connected with the COFs at the macro-level as shown in the TEM images.

3. The authors claimed that "Additionally, the increase of the length of NRs from 12 to 40 nm would increase the intensity of the photocurrents, suggesting that a longer NR promotes the photo-excited charge separation efficiency". However, some reported works suggest that smaller particles would inhibit the recombination of the charge carriers (Photocatalytic Conversion of CO₂ into Renewable Hydrocarbon Fuels: State-of-the-Art Accomplishment, Challenges, and Prospects, *Adv. Mater.* 2014, 26, 4607.). Can the authors explain about this conclusion?

4. Poor linear relationship between the time and gas generation rate can be observed in the Figure 5a,b. Any explanations?

5. The catalyst is used in gas-solid system for CO₂ photoreduction only with water vapor without any sacrificial agents, which means another half-reaction (water oxidation) of the CO₂ reduction also happens in this system. Therefore, O₂ generation rate is highly suggested to measured to prove the validity of this reduction system.

Reviewer #3 (Remarks to the Author):

This is an exciting manuscript on the topic of "Double shelled hollow superstructures from self-assembled² CdSe/CdS nanorods within covalent organic frameworks³ for efficient CO₂ photoreduction". This manuscript has been written well, and the logic behind this work is sound. Hence, I recommend acceptance per minor revision.

1. It seems that the CO₂ to CO formation mechanism is not well established. What is the role of the COF, and what is the part of CdSe/CdS nanorods?

2. I think some theoretical calculation would be nice to provide this mechanism.

3. Also, it would be nice to perform an isotope labeling experiment to prove the course of the CO₂ would be nice.

4. The authors mentioned that "the self-assembled porous nanocomposites exhibit remarkably higher activity ($\approx 65.8 \mu\text{mol g}^{-1}\text{h}^{-1}$) toward CO₂ to CO in solid-gas regime". Can they provide a list of reported CO₂ reduction catalysts and compare their results?

5. The BET surface area looks very modest. I am not so sure about the diffusion of CO₂ to the catalytic site. This should be explained carefully.

6. There has been a good number of literature on COF nanospheres and their behaviors. I would recommend the authors to take a look [J. Am. Chem. Soc., 2021, 143, 8426; J. Am. Chem. Soc., 2021, 143, 955; J. Am. Chem. Soc., 2019, 141, 20371; ACS Nano, 2021, 15, 12723; J. Am. Chem. Soc., 2021, 143, 20916]

Reviewer #4 (Remarks to the Author):

The authors in this manuscript fabricated a micro-meso hierarchically porous

nanocomposite with CdSe/CdS nanorods and imine-based polymers, which possessed double-shelled hollow structures effective for CO₂-to-CO photoreduction. The hollow structure was characterized in detail. I do have concerns on this manuscript that deter me from recommending its publication, and my questions are as follows:

1. Structural evidence must be shown for claiming "COF", since it is not a random name. The authors need to show the crystallinity from X-ray measurements, like GIWAXS, otherwise the organic shell can be only defined as Porous Framework.
2. In Figure 1, the construction of hollow structures requires fairly harsh conditions with high AcOH concentration up to 60 vol%. While the condition is compatible with TAPT-DMTA-H, I doubt whether such acidities cause structural, compositional or spectral damage to CdSe/CdS nanorods? Another concern about the acidity is that since hollow structures are formed under acidic conditions, do they remain stable under basic conditions, as CO₂ reduction is anticipated to consume protons and cause local basicity to some extent?
3. The solid-state UV-Vis absorption spectra of all the samples in Figure 2 show absorption maximum at exactly 1, is it a coincidence? If spectra were normalized, the information has to be mentioned either in the figure or in the caption.
4. It was observed that "the exciton lifetime of the TAPT-DMTA/NR40-H was markedly shortened from 63.06 ns of the NR40 to 0.22 ns". For such a dramatic decline of exciton lifetime, although enhanced charge separation makes contribution, I doubt other reasons can also exist, as in Figure 3c and 3d, the CdSe/CdS NRs are in very tight and condense bundles. In this situation, will self-quenching of adjacent CdSe/CdS NRs also shorten the exciton lifetime? (This self-quenching may not be that helpful to deliver photogenerated electrons for CO₂ reduction)
5. The mutual interaction between QDs and COF was only characterized by steady-state and time resolved photoluminescence. I agree with the author that the quenched lifetime indicates the strong interaction between the excited states of QDs and the COF. However, we might be able to distinguish electron transfer or hole transfer. The mechanism as depicted in Figure 4c corresponds to the type II charge separation in heterojunction structure, which should show the slower recombination. Could the author explain a little more about the mechanism?
6. The authors also performed CO₂ reduction in the absence of sacrificial agents and observed O₂. Does evolved O₂ follow the expected stoichiometry to CO?
7. The data statistics can be only seen in transient photocurrent measurements. However, most of catalytic results, which are the key points of the manuscript, were presented with single measurement. It is very disappointing, given the high expectation on the impact of the manuscript.
8. The quantum yield is an important index to evaluate photocatalytic efficiency. The author should give the catalytic quantum yield of the tests.

Response to reviewers' comments

Reviewer #1:

Dr. Wei and co-workers synthesized COF-based nanorod composites and fully characterized them. One of the COF composites in the series shows excellent photocatalytic CO₂ reduction under the gas phase without any sacrificial agents. I recommend publishing this work after a major revision. Below are my comments and suggestion.

Response: We thank this reviewer for the positive comment on our present manuscript, and hope our revision will clarify his/her concerns.

1) Lines 89, 90, "CdSe/CdS coated with octadecylphosphonic acid (ODPA) and hexylphosphonic acid (HPA) ligands via a seeded growth approach", can the authors elaborate more on the role of ODPA and HPA?

Response: Thanks for the comment. We have synthesized CdSe/CdS nanorods according to a reported approach developed by Manna et al. (Nano Lett. 2007, 7, 2942). The synthesis was based on the co-injection of appropriate precursors and preformed spherical CdSe nanocrystal seeds (nearly monodisperse in size) in a reaction flask that contains a mixture of hot surfactants (CdO-TOPO/ODPA/HPA mixture) for the anisotropic growth of CdS nanocrystals. Immediately after injection, CdS started growing preferentially on the CdSe seeds rather than forming separate nuclei in solution because the activation energy for heterogeneous nucleation was much lower than that for homogeneous nucleation. As the homogeneous nucleation was bypassed by the presence of the seeds, all nanocrystals underwent almost identical growth conditions and therefore they maintained a narrow distribution of lengths and diameters during their evolution. During the growth process, the addition of phosphonic acid is essential to the formation of anisotropic CdSe/CdS nanorods (Chem. Mater. 2007, 19, 2573). Moreover, a mixture of phosphonic acid differing by the alkyl chain lengths is also crucial to achieve the length control of the nanorods. Herman and coworkers found that a single phosphonic acid (like octadecylphosphonic acid, ODPA) would yield CdSe nanorods with low aspect ratios. On the contrary, a mixture of ODPA and hexylphosphonic acid (HPA) could lead to CdSe nanorods with large aspect ratios. Hence, the HPA ligand mainly plays a role in terminating the growth in the $\langle 100 \rangle$ direction, which consequently leads to the elongation of the nanorods along the $\langle 001 \rangle$ direction.

Overall, both the ODPA and HPA ligands could bond to the surface of CdSe/CdS nanorods strongly, which render these nanorods dispersible in nonpolar solvents, like hexane, toluene, or CHCl₃. Moreover, as selective surfactant, HPA terminated the growth in the $\langle 100 \rangle$ direction, and the preferential heterogeneous nucleation and anisotropic growth of c-axis elongated wurtzite CdS onto the primary CdSe seeds ultimately resulted in CdSe/CdS core-shell nanorods. This discussion has been partly included into the revised manuscript.

2) Lines 106, 107, what is the role of surfactant DTAB?

Response: Thank for the comment. Although it is well known that THF is miscible with water, adding other nonpolar solutes into the THF, such as monomers of COFs (TAPT and DMTA) and CdSe/CdS nanorods, would lead to a biphasic structure. In this regard, we added the cationic surfactant DTAB into the mixture as emulsifier, which were able to produce compartmentalized THF droplets dispersed in water after emulsification. These micrometer-sized oil droplets dispersed in water served as microreactors for the polycondensation reactions, which proceeded rapidly catalyzed by the Lewis acid- $\text{Sc}(\text{OTf})_3$. Meanwhile, the evaporation of the THF phase would trigger the condensation followed by the assembly of nanorods within the droplets. After the complete removal of the THF phase, TAPT-DMTA/NR hybridized colloids were formed, and their surfaces were covered by a layer of DTAB, driven by the hydrophobic interactions, which render these colloids dispersible in water.

Additionally, we have performed a control experiment without the addition of the surfactant DTAB, while keeping other conditions identical to the preparation procedure. The result showed that the particles are rather polydisperse in size, and the CdSe/CdS nanorods are not distributed within the particles homogeneously (**Figure R1**). Hence, we conclude that the DTAB plays an important role in producing TAPT-DMTA/NR hybridized colloids. This part has been included into Supplementary Information

Figure R1. TEM images of nanocomposites of TAPB-DMTA/NR40 prepared in the absence of DTAB.

3) $\text{Sc}(\text{OTf})_3$ plays as a Lewis catalyst. Why do the authors have to use AcOH 40% and 60% to catalyze the imine-linked COFs formation?

Response: Thanks for the comment. We used a two-step polycondensation-crystallization approach to produce COFs colloids. In the first step, we exploited $\text{Sc}(\text{OTf})_3$ as Lewis catalyst, which catalyzes the formation of imine bond between TAPT and DMTA rapidly, thereby producing amorphous polymers with low BET surface area ($S_{\text{BET}} = 44.5 \text{ m}^2 \text{ g}^{-1}$). In the second step, we converted the amorphous polymer into crystallized polymer TAPT-DMTA COFs in the presence of AcOH differing by the concentration, because of the dynamic covalent bonding of imine. After crystallization, the BET surface area increased to $S_{\text{BET}} = 559.6 \text{ m}^2 \text{ g}^{-1}$.

In the second crystallization step, we have done control experiments by replacing AcOH with Sc(OTf)₃. However, the BET surface area of the polymer did not increase apparently, indicating the unsuccessful crystallization of TAPT-DMTA polymer under Sc(OTf)₃. This can be explained as follows: 1) The surface of the TAPT-DMTA polymer is coated with cationic surfactant DTAB, which gives rise to the positive zeta potential of the TAPT-DMTA polymer. The formation electric double layer would screen the positive charged Sr³⁺ ions and hampers the diffusion of Sr³⁺ to the TAPT-DMTA polymer, which results in rather low crystallization rate. On the contrary, the weak acid AcOH could access to the TAPT-DMTA polymer more easily and thereby affect the subsequent crystallization. 2) Concerning on the molecular structure of the TAPT-DMTA polymer, it is hydrophobic and rejects the metallic cations. While the acetyl group of the AcOH is hydrophobic to some extent, which permits their diffusion into the TAPT-DMTA polymers and enables the crystallization of the polymers.

4) The authors should double-check the inner and outer shell thicknesses in TAPT-DMTA/NR40-H. If based on Fig 9f, I don't think it is 8 and 30 nm, respectively.

Response: We sincerely thank this reviewer for pointing this issue out. We have double checked the shell thickness of both the inner shell and the outer shell of TAPT-DMTA/NR40-H sample. We have counted more than 200 particles to determine the thickness of the inner and outer shells. The mean shell thickness of the outer shell is measured to be 29.5 ± 3.5 nm. While the inner shell is rather inhomogeneous, and the thickness is polydisperse with a value of 14.6 ± 3.4 nm. We have modified the values in the revised manuscript. We thank this reviewer again for improving the precision of such data.

Figure R2. (a) The TEM image of TAPT-DMTA/NR40-H, illustrating the shell thicknesses of both the outer and inner shells. Size histograms of the thicknesses of the outer shell (b) and the inner shell (c). Each histogram was made by counting more than 200 particles from TEM images.

5) The authors mentioned high CO₂ uptake in the introduction. However, I don't think the CO₂ uptake of the

COF composite materials in this work is high (the best one in the series is just 13.8 cc/g). Therefore, it is not convincing to say, “indicating that the as-formed meso-micro hierarchical porous structure promotes the adsorption of CO₂.” (line 161).

Response: Thanks for the comment. We agree with this reviewer that the CO₂ uptake of the present photocatalyst is modest when it is compared to other porous materials. We made the statement “the as-formed meso-micro hierarchical porous structure promotes the adsorption of CO₂” based on the observation as follows: 1) The CO₂ uptake of **TAPT-DMTA/NR40-H** (13.8 cm³ g⁻¹) is higher than that of the **TAPT-DMTA-H** (9.2 cm³ g⁻¹) in the absence of NR40; 2) The CO₂ uptake of **TAPT-DMTA/NR40-H** (13.8 cm³ g⁻¹) is much higher than that of the solid counterpart (5.3 cm³ g⁻¹). Considering that the **TAPT-DMTA-H** sample mainly consists of micropores, whereas the **TAPT-DMTA/NR40-H** consists of both micropores and mesopores, determined from the nitrogen sorption experiments, we conclude that the as-formed meso-micro hierarchical porous structure could promote the adsorption of CO₂.

6) The authors should elaborate more on how to determine conduction band based on the flat band and bandgap energy.

Response: Thanks for the comment. Mott-Schottky measurements are applied to exploit the semiconducting properties and the electronic band positions.

The flat band potentials of the semiconducting materials could be determined by the Mott-Schottky plot from the electrochemical approach (which could be found in the experimental section). The Mott-Schottky equation is given as follows:

$$\frac{1}{C^2} = \frac{2}{\epsilon\epsilon_0 A^2 e N_D} \left(V - V_{fb} - \frac{k_B T}{e} \right) \quad (R1)$$

Where C and A are the interfacial capacitance and area, respectively, V the applied voltage, N_D the number of donors, k_B is Boltzmann’s constant, T the absolute temperature, and e is the electronic charge. Hence, a plot of $1/C^2$ against V should result in a straight line from which the flat band potential can be determined from the intercept on the V axis. The value of N_D could be determined from the slope knowing ϵ and A . The C is determined by electrochemical impedance spectroscopy (EIS). Since the semiconductor capacitance could be extracted from the complex component of the measured impedance, Z'' , as a function of the angular frequency (ω) of the voltage:

$$Z'' = \frac{j}{\omega C} \quad (R2)$$

We applied several different angular frequencies of the voltage to determine the V_{fb} . All the CdSe, CdS NRs and TAPT-DMTA exhibited positive slopes in the Mott-Schottky plots, a characteristic typical of a n -type semiconductor. In addition, we used the Ag/AgCl as the reference electrode, thus the V_{fb} vs. NHE ($V_{fb}(\text{NHE})$) could be determined by the following equation:

$$V_{fb}(\text{NHE}) = V_{fb}(\text{Ag/AgCl}) + 0.199 \text{ V} \quad (R3)$$

Therefore, the flat band energy for CdSe, CdSe/CdS (NR40), and TAPT-DMTA-H is determined to be -0.94, -1.17 and -0.85 eV, respectively. In general, the bottom of conduction band or LUMO stays very close to the flat band potential in *n*-type semiconductors (Nat. Commun. 2019, 10, 2467; J. Am. Chem. Soc. 2020, 142, 9752–9762). To a first approximation, we used the flat band energy as the conduction band. The HOMO level or valence band could be calculated by considering the optical band gap.

Figure R3. Mott-Schottky plots of CdSe seeds (a), CdSe/CdS NR40 (b), and TAPT-DMTA-H (c). The blue, red and black curves were measured by applying a frequency of 2000, 1500, and 1000 Hz, respectively. The insets in the plots are the mathematical linear equations.

7) Can the authors explain a bit more on Fig. 4 e and f? How to interpret to get 63.06 and 0.22 ns?

Response: Thanks for the comment. We used the τ_{ave} to show the PL lifetime of the samples. For example, the NR40 exhibited triple exponent decay with time constants of $\tau_1 = 11.13$ ns (58.57%), $\tau_2 = 45.23$ ns (27.16%) and $\tau_3 = 310.13$ ns (14.27%), and the calculated average time of $\tau_{ave} = 63.06$ ns. In the revised manuscript, we also measured the NR40 assemblies prepared by a similar emulsion confined assembly method, which is suggested by Reviewer #4, and the τ_{ave} of the NR40 assemblies is 39.96 ns, such a decrease in the exciton lifetime τ_{ave} from NR40 to NR40 assemblies could be due to the self-quenching in self-assembled NR assemblies. The TAPT-DMTA/NR40-H exhibited triple exponent decay with time constants of $\tau_1 = 0.03$ ns (57.84%), $\tau_2 = 0.28$ ns (29.40%) and $\tau_3 = 0.96$ ns (12.77%), and the calculated average time of $\tau_{ave} = 0.22$ ns. These data were included in Table S2 in the Supplementary Information.

Figure R4. Time resolved PL spectra of (a) NR40, (b) NR40 assemblies, and (c) TAPT-DMTA/NR40-H.

8) *It is useful to readers if the authors explain why the longer length of NR will promote the photoexcited charge separation efficiently?*

Response: Thanks for the comment. The 1D CdSe/CdS dot-in-rod is a typical Type-I heterojunction, where the conduction band (CB) and valence band (VB) of CdS NRs straddle the narrower band gap of CdSe core. The CdS NRs act as light harvesting antenna to strongly absorb light and to transport photo excitation energy into the CdSe core where light emission or photochemical reactions can occur. In the present manuscript, we have introduced TAPT-DMTA-COFs, which enables the formation of Type-II heterojunction with CdS, and it could be served as electron acceptors based on the CB energy level of CdS and TAPT-DMTA (**Figure R5a**). Hence, a donor (CdSe)-absorber (CdS)-acceptor/catalyst (TAPT-DMTA) inorganic/organic nanocomposite superstructure is constructed for direct CO₂ photoreduction. Therefore, the distance of charge separation in the charge separated state (CdSe⁺-CdS-TAPT-DMTA⁻) can be simply controlled by the CdS rod length (**Figure R5b**). Such heterojunction design is akin to the previous report on Pt-tip coated CdSe/CdS dot-in-rod structure, where the CdSe, CdS, and Pt serves as donor, absorber, and acceptor, respectively (Chem. Soc. Rev. 2016, 45, 3781). The distance of charge separation of CdSe/CdS/Pt is strongly correlated with the CdS rod length.

Figure R5. (a) energy level diagram of CdSe, CdS, and TAPT-DMTA in the photocatalytic system on the NHE scale. (b) Representation of the CO_2 photoreduction in the presence of the present CdSe-CdS-TAPT-DMTA heterojunctions. e^-h^+ represents the photo-generated excitonic state in the CdS domain.

9) Line 225, the authors should consider the CO_2 uptake as discussed above.

Response: Thanks for the comment. We have measured the CO_2 uptake for the samples differing by the length of the NRs. The CO_2 uptake for the double shelled superstructures of TAPT/DMTA/NR12-H and TAPT/DMTA/NR24-H is 10.4 and 12.4 $\text{cm}^3 \text{g}^{-1}$, respectively, which is very close to that of the TAPT/DMTA/NR40-H (13.8 $\text{cm}^3 \text{g}^{-1}$). Nevertheless, the CO_2 uptake for all the double shelled superstructures is much higher than that of the solid one.

Based on the CO_2 uptake data, we speculate that the slight difference of the CO_2 uptake volume among various double shelled samples could have negligible impact on the CO_2 photoreduction reaction. In other words, the distinct activity in CO_2 photoreduction in TAPT/DMTA/NR12-H, TAPT/DMTA/NR24-H, and TAPT/DMTA/NR40-H could be attributed to their distinct photoexcited charge separation efficiency, which is highly associated with the length of CdS NRs as discussed in response to Q8.

We have included the CO_2 uptake data into the Supplementary Information.

Figure R6. (a) the CO₂ adsorption curves of various samples at 298 K (b) the summarized CO₂ adsorption values.

10) “Denoted as” is just fine, but I would use “denoted by”. Also, lines 125, 183, and 187: it should be “replaced with” instead of “replaced by”.

Response: Thanks for the suggestion. “Denoted by” instead of “denoted as” and “replaced with” instead of “replaced by” are used in the revised manuscript.

11) I think the catalytic performance of TAPT-DMTA/NR40-H starts dropping from the third cycle. The authors should explain and revise the manuscript accordingly.

Response: Thanks for the comment. Indeed, the catalytic performance of TAPT-DMTA/NR40-H starts dropping from the third cycle, which is probably due to the formation of surface products and the lack of any catalyst regeneration process. We have noted this point in the revised manuscript.

The revision is as follows: “We note that a slight decrease in the activity after the third cycle could be observed, which is probably due to the formation of surface products and the lack of any catalyst regeneration process.”

12) I suggest the author cite relevant reviews about using COFs for CO₂ photoreduction recently published by Nguyen in ACS Catalyst or Advanced Energy Materials (ACS Catal. 2021, 11, 9809–9824; Adv. Energy Mater. 2020 10, 2002091).

Response: Thanks for the suggestion. We have read these reviews carefully and these reviews provide impressive insights for understanding the CO₂ photoreduction using COFs. These literatures have been cited as Ref. 34 and 35 in the introduction part.

Reviewer #2

In this work, the authors demonstrate the design and fabrication of integrated organic and inorganic semiconductors for CO₂ photoreduction. The authors incorporate self-assembled CdSe/CdS dot-in-rod nanorods within the micro-meso porous COFs matrix, resulting in a double-shelled inorganic/organic nanocomposites. In addition, the authors investigate how the length of the CdSe/CdS nanorods effects the long distance separation of photo-induced electrons and holes. However, the authors didn't clearly validate the interaction between the CdSe/CdS nanorods and COFs matrix with related measurements. The gas-solid system of the CO₂ photoreduction reaction in this manuscript also needs more proof of the complete reactions. Overall, the claims are poorly supported by solid evidence. Therefore, I would not recommend it to be published in this journal. The following are some suggestions for the authors to improve the quality of this paper for more-specialized journals.

Response: We thank this reviewer for the critical comment on our present manuscript, and hope our revision will clarify his/her concerns.

1. Self-assembly of the CdSe/CdS dot-in-rod nanorods have been widely studied in previous works(Synthesis and micrometer-scale assembly of colloidal CdSe/CdS nanorods prepared by a seeded growth approach, Nano letters, 2007, 7, 2942; CdSe and CdSe/CdS Nanorod Solids, J. Am. Chem. Soc. 2004, 126, 12984), and the structure of the TAPA/TAPB/TAPT-DMTA COFs used in this work is also examined explicitly (Stable, crystalline, porous, covalent organic frameworks as a platform for chiral organocatalysts. Nature Chemistry, 2015, 7, 905; Using sound to synthesize covalent organic frameworks in water, Nature Synthesis, 2022, 1, 87). But the authors combine these two semiconductors together without highlighting the interaction between these two components.

Response: Thanks for the comment. We agree with this reviewer that the building blocks in the present work is not new—self-assembly of CdSe/CdS dot-in-rod nanorods into hexagonal close-packed superlattices has been reported by Talapin, Murray, Manna and others, synthesis of TAPT-DMTA COFs has also been extensively studied. In the present work, our primary goal was to develop a method to combine the semiconductors from structurally dissimilar inorganic and organic materials, thereby giving rise to nanocomposites with several remarkable features including (1) The contact between CdSe/CdS dot-in-rod nanorods and TAPT-DMTA COFs enables to build the heterojunctions. In the CdSe/CdS dot-in-rod model, CdS NRs act as light harvesting antenna to strongly absorb light and to transport photo excitation energy into the CdSe core where light emission or photochemical reactions can occur. The TAPT-DMTA COFs could be served as electron acceptor on the basis of the energy level of CdSe, CdS, and TAPT-DMTA. Hence, a donor (CdSe)-absorber (CdS)-acceptor/catalyst (TAPT-DMTA) inorganic/organic nanocomposite superstructure is constructed for CO₂ photoreduction. (2) The porous structure of TAPT-DMTA COFs allows the

uptake/diffusion of CO₂ to the catalytically active sites. (3) The double shelled structure of the nanocomposites enables the strong scattering or diffusing effect when photons transmit through the sample, thereby leading to enhanced the light absorption compared to the solid counterpart.

2. *The authors claim that “The inner shell is comprised of self-assembled NRs that is in close contact with the outer shell of the COFs, which potentially facilitates the photogenerated electron transfer from NRs to the backbones of the COFs.” However, the NRs do not atomically interact with COFs, and are only partially connected with the COFs at the macro-level as shown in the TEM images.*

Response: Thanks for the comment. We have conducted the high-resolution electron microscopy to understand the interactions between COFs and CdSe/CdS NRs. First, the lattice fringe of the nanorods could be clearly observed, which could be assigned to the (0002) crystal plane of the CdS, in agreement with a fast growth rate of the nanorods along the *c*-axis of their wurtzite structure. Additionally, the hexagonal prism structure of the NRs could be observed from one tip-end aligned with the electron beam inside the TEM (**Figure R1**). Second, the lamellar structure of the TAPT-DMTA COFs could be observed at the edge of the shell, despite the fact of poor crystallinity of the COFs. Third, no apparent gap between TAPT-DMTA COFs and CdSe/CdS NRs could be observed from the magnified TEM image, revealing that these two semiconductors are contacted closely (**Figure R1e**). Nevertheless, we could not claim that these NRs are interacted with COFs atomically, because of their large lattice mismatch—the lattice fringe for CdS (0002) is 0.34 nm, whereas the length of the repeating unit for the TAPT-DMTA COFs is calculated to be ~2.5 nm, one order of magnitude larger than that of the CdS. Therefore, they are contacted physically through van der Waals interactions, which is the most probable here.

Additionally, we note that these CdSe/CdS NRs could be fully contacted with the TAPT-DMTA COFs when these NRs are covered with a solid TAPT-DMTA shell. In the manuscript, we have also prepared such a sample, denoted by **TAPT-DMTA/NR40-S**. However, the **TAPT-DMTA/NR40-S** sample with a solid shell shows markedly lower CO₂ uptake and reduced light absorption than that of the hollow counterpart **TAPT-DMTA/NR40-H**. Consequently, the **TAPT-DMTA/NR40-S** shows a lower CO formation rate (~17 μmol g⁻¹ h⁻¹) during the CO₂ photoreduction experiment. Hence, we conclude that both the double shelled structures with hollow cavities and the contact between NRs and COFs are key elements in improving the activity of the CO₂ photoreduction.

Figure R1. HRTEM images of nanocomposites of TAPT-DMTA/NR40-H. (a-c) low magnification TEM images showing the contact between the inner shell of NRs assemblies and the outer shell of TAPT-DMTA polymers; (d-e) high magnification TEM images showing the lattice fringes of the NRs and the boundary between NR assemblies and TAPT-DMTA polymers.

3. The authors claimed that “Additionally, the increase of the length of NRs from 12 to 40 nm would increase the intensity of the photocurrents, suggesting that a longer NR promotes the photo-excited charge separation efficiency”. However, some reported works suggest that smaller particles would inhibit the recombination of the charge carriers (*Photocatalytic Conversion of CO₂ into Renewable Hydrocarbon Fuels: State-of-the-Art Accomplishment, Challenges, and Prospects, Adv. Mater.* 2014, 26, 4607.). Can the authors explain about this conclusion?

Response: Thanks for the comment. In zero-dimensional (0D) materials, like quantum dots (QDs), the small size limits the light absorption cross section (scaled with their volume) and the distance for charge separations (scaled with their diameters) (*J. Chem. Phys.*, 1984, 80, 4403). In the 1D CdSe/CdS dot-in-rod model, CdS NRs act as light harvesting antenna to strongly absorb light and to transport photo excitation energy into the CdSe core where light emission or photochemical reactions can occur. In the present manuscript, we have introduced TAPT-DMTA-COFs, which enables the formation of Type-II heterojunction with CdS, and it could be served as electron acceptors based on the CB energy level of CdS and TAPT-DMTA (**Figure R2**). Hence, a donor (CdSe)-absorber (CdS)-acceptor/catalyst (TAPT-DMTA) inorganic/organic nanocomposite superstructure is constructed for direct CO₂ photoreduction. Therefore, the distance of charge separation in

the charge separated state ($\text{CdSe}^+-\text{CdS}-\text{TAPT-DMTA}^-$) can be simply controlled by the CdS rod length. Such heterojunction design is akin to the previous report on Pt-tip coated CdSe/CdS dot-in-rod structure, where the CdSe, CdS, and Pt serves as donor, absorber, and acceptor, respectively (Chem. Soc. Rev. 2016, 45, 3781). The distance of charge separation of CdSe/CdS/Pt is strongly correlated with the CdS rod length. We note that not all the tips of CdS/CdSe NRs were contacted with the shell of the TAPT-DMTA COFs in the nanocomposites, and a certain portion of the CdS/CdSe NRs were contacted with TAPT-DMTA COFs through their lateral facets, which may reduce the charge separation efficiency. In an ideal scenario, all the CdSe/CdS NRs were contacted with the shell through their tip-ends, which consequently yield the vertical assembly of CdSe/CdS NRs inside the hollow shell of TAPT-DMTA COFs, thereby maximizing the photogenerated charge separation efficiency. However, in the assembly experiment, the aliphatic chain-chain van der Waals interaction from the lateral facets of NRs is rather strong, which leads to the assembly of CdSe/CdS NRs in a side-by-side fashion and does not allow that all the tip-ends of the NRs are contacted with the shell of TAPT-DMTA.

Overall, on the basis of the photocurrent experiments and the PL lifetime measurements of various nanocomposites differing by the length of CdSe/CdS NRs, it is highly plausible that the longer CdS NR could lead to the more efficient photogenerated charge separation in the presence of TAPT-DMTA COFs.

Figure R2. (a) energy level diagram of CdSe, CdS, and TAPT-DMTA in the photocatalytic system on the NHE scale. (b) Representation of the CO_2 photoreduction in the presence of the present CdSe-CdS-TAPT-DMTA heterojunctions. e^-h^+ represents the photo-generated excitonic state in the CdS domain.

4. Poor linear relationship between the time and gas generation rate can be observed in the Figure 5a,b. Any explanations?

Response: Thanks for the comment. We believe that the poor linear relationship could be due to the initial gas uptake-release from the porous photocatalysts. To confirm this argument, we have performed the long-time running of the photoreduction experiments. The result shows that more satisfied linear relationship could be observed from a 36 h photoreduction reaction.

Figure R3. Time-dependent CO generation in the presence of 50 mg of TAPT-DMTA/NR40-H.

5. The catalyst is used in gas-solid system for CO₂ photoreduction only with water vapor without any sacrificial agents, which means another half-reaction (water oxidation) of the CO₂ reduction also happens in this system. Therefore, O₂ generation rate is highly suggested to measure to prove the validity of this reduction system.

Response: Thanks for the comment. We have performed the CO₂ photoreduction without any sacrificial agents. To confirm another half reaction, we have detected the oxygen evolution reaction by GC. First of all, we have conducted a control experiment done without the addition water, and the result showed that no detectable CO could be observed for TAPT-DMTA/NR40-H, indicating that the CO₂ photoreduction cannot proceed in the absence of water. Hence, a most probable half reaction is the water oxidation: $2\text{H}_2\text{O} \rightarrow \text{O}_2 + 4\text{H}^+ + 4\text{e}^-$. We used 50 mg of the TAPT-DMTA/NR40-H photocatalyst to conduct the CO₂ photoreduction, and the products were detected by GC. The standard curve for oxygen gas was first depicted by GC, which was applied to quantify the amount of oxygen generated during the water oxidation reaction. The result in **Figure R4** revealed that the stoichiometry between CO and O₂ gas is measured to be around 2:1 at three different reaction times. Hence, we conclude that another half reaction is the oxygen evolution from water

oxidation reaction.

Figure R4. (a) Standard curve of the oxygen gas determined by GC. (b) The quantity of CO and O₂ catalyzed by 50 mg of the TAPT-DMTA/NR40-H photocatalyst at different reaction times.

Reviewer #3:

This is an exciting manuscript on the topic of "Double shelled hollow superstructures from self-assembled CdSe/CdS nanorods within covalent organic frameworks for efficient CO₂ photoreduction". This manuscript has been written well, and the logic behind this work is sound. Hence, I recommend acceptance per minor revision.

Response: We thank this reviewer for the positive comment on our present manuscript, and hope our revision will clarify his/her concerns.

1. It seems that the CO₂ to CO formation mechanism is not well established. What is the role of the COF, and what is the part of CdSe/CdS nanorods?

Response: Thanks for the comment. Concerning the role of CdSe/CdS nanorods in the photoreduction of CO₂ to CO, the small diameter (~ 3 nm) maintains quantum confinement effects of the exciton motion in the radial direction, whereas the axial length direction enables the bulk large absorption cross section and the long-distance charge separation. For the CdSe/CdS nanorods, CdS nanorods act as light harvesting antenna to strongly absorb light and to transport photo excitation energy into the CdSe core where light emission or photochemical reactions can occur. Here, we have introduced TAPT-DMTA-COFs, which enables the formation of Type-II heterojunction with CdS, and it could be served as electron acceptors based on the CB energy level of CdS and TAPT-DMTA. Hence, a donor (CdSe)-absorber (CdS)-acceptor/catalyst (TAPT-DMTA) inorganic/organic nanocomposite superstructure is constructed for direct CO₂ photoreduction. Hence, the distance of charge separation in the charge separated state (CdSe⁺-CdS-TAPT-DMTA⁻) can be simply controlled by the CdS rod length. Such heterojunction design is akin to the previous report on Pt-tip coated CdSe/CdS dot-in-rod structure, where the CdSe, CdS, and Pt serves as donor, absorber, and acceptor, respectively (Chem. Soc. Rev. 2016, 45, 3781). The distance of charge separation of CdSe/CdS/Pt is strongly correlated with the CdS rod length.

2. I think some theoretical calculation would be nice to provide this mechanism.

Response: Thanks for the suggestion. We have mainly analyzed the energy levels of the various semiconductors in the present work. Based on the energy level of the TAPT-DMTA COFs, we believe that CO₂ photoreduction takes place at the COFs. To further identify the active site of the TAPT-DMTA COFs, we performed the density functional theory (DFT) calculations on the unit of the TAPT-DMTA COFs, which

gives the information of frontier molecular orbitals, HOMO-LUMO energy level and the molecular electrostatic potentials. We found that the more negative electrostatic potential could be found in the imine and triazine positions, suggesting that these positions are most probably catalytically active sites for the CO₂ reduction reaction. We have included this part in the Supplementary information.

Figure R1. (a) Frontier molecular orbitals and HOMO-LUMO energy level of the unit of TAPT-DMTA-COFs, (b) the molecular electrostatic potential map of the unit of TAPT-DMTA-COFs.

3. Also, it would be nice to perform an isotope labeling experiment to prove the course of the CO₂ would be nice.

Response: Thanks for the suggestion. We have performed the isotope labeling experiment by using ¹³CO₂ as the source of carbon dioxide. After photoreduction under identical conditions as described in the experimental section, the gas products were analyzed by the GC-MS. The result revealed that ¹³CO could be observed in the mass spectra at $m/z = 29$, confirming that the generated CO was originated from the photoreduction of CO₂.

Figure R2. GC–MS analysis of ¹³CO₂ isotope after ¹³CO₂ photoreduction in the presence of TAPT-DMTA/NR40-H.

4. The authors mentioned that "the self-assembled porous nanocomposites exhibit remarkably higher activity ($\approx 65.8 \mu\text{mol g}^{-1} \text{h}^{-1}$) toward CO₂ to CO in solid-gas regime". Can they provide a list of reported CO₂ reduction catalysts and compare their results?

Response: Thanks for the comment, we have provided a list of reported catalysts for CO₂ photoreduction in gas-solid regime in the last four years, which is summarized in Table R1. The Table R1 shows that the CO generation rate in most of the reported photocatalysts, in terms of $\mu\text{mol g}^{-1} \text{h}^{-1}$, is much lower than that of the present work.

Table R1. Comparison of photocatalytic activity with reported CO₂ to CO conversion rate based on the gas-solid systems.

No.	Light Source	Photocatalyst	CO ($\mu\text{mol g}^{-1} \text{h}^{-1}$)	Reference
[1]	300W Xe lamp, $\lambda > 420 \text{ nm}$	BiVO ₄ /C/Cu ₂ O	3.0	ACS Catal. 2018 ⁴
[2]	300W Xe lamp	Cu/CeO _{2-x}	1.7	ACS Catal. 2019 ⁵
[3]	300W Xe lamp	BiOIO ₃	17.3	Adv. Mater. 2020 ⁶
[4]	Xe lamp, $\lambda > 800 \text{ nm}$	CuS	14.5	J. Am. Chem. Soc. 2019 ⁷
[5]	300W Xe lamp	BiOIO ₃	5.4	Adv. Funct. Mater. 2018 ⁸
[6]	300W Xe lamp	Bi ₂ O ₂ (OH)(NO ₃)	8.1	Adv. Mater. 2019 ⁹
[7]	UV, 305 nm	Cs ₃ Bi ₂ I ₉	7.8	J. Am. Chem. Soc. 2019 ¹⁰

[8]	150 W Xe lamp	TiO ₂ /g-C ₃ N ₄	2.0	Appl. Catal. B 2019 ¹¹
[9]	300W Xe lamp	Li ₂ TiO ₃ /TiO ₂	1.7	Appl. Catal. B 2020 ¹²
[10]	300W Xe lamp, λ>400 nm	SnS ₂ /SnO ₂ HoMSs	4.0	Angew. Chem. Int. Ed. 2020 ¹³
[11]	300W Xe lamp	Ni-SA-x/ZrO ₂	11.8	Adv. Energy Mater. 2020 ¹⁴
[12]	300W Xe lamp	red phosphorus	2.1	J. Mater. Chem. A 2021 ¹⁵
[13]	300W Xe lamp	Ti ₃ C ₂ MXene/ g-C ₃ N ₄	5.2	Appl. Catal. B 2020 ¹⁶
[14]	300W Xe lamp	g-C ₃ N ₄ /Bi ₂ WO ₆	5.2	J. Mater. Chem. A 2015 ¹⁷
[15]	400 nm LEDs	ZnSe/CdS DOR	11.3	Adv. Mater. 2021 ¹⁸
[16]	Xenon lamp (200-1000 nm)	TAPBB-COF	24.6	ChemSusChem 2020 ¹⁹
[17]	300W Xe lamp (380-800 nm)	COF-318-TiO ₂	69.67	Angew. Chem. Int. Ed. 2020 ²⁰
[18]	300W Xe lamp (300-1200 nm)	MTCN-H (ys)	16.87	Adv. Mater. 2021 ²¹
[19]	200W Xe lamp (AM 1.5)	QS-Co ₃ O ₄ (ZIF-67)	46.3	J. Am. Chem. Soc. 2019 ²²
[20]	300W Xe lamp, λ>420 nm	TAPT-DMTA/NR40-H	64.6	This work

5. The BET surface area looks very modest. I am not so sure about the diffusion of CO₂ to the catalytic site. This should be explained carefully.

Response: Thanks for the comment. The BET surface area of the present double shelled hollow structure is not high when it is compared to COFs prepared under other conditions. Nevertheless, we believe that the diffusion of CO₂ to the catalytic site is not hampered, because the CO₂ uptake experiment shows that a higher CO₂ uptake of 13.8 cm³ g⁻¹ was observed for TAPT-DMTA/NR40-H than that of the TAPT-DMTA-H in the absence of the inner shell of NRs assemblies (9.2 cm³ g⁻¹). Additionally, on the basis of the energy level of CdSe/CdS/TAPT-DMTA donor/absorber/acceptor, the CO₂ reduction reaction takes place at the TAPT-DMTA COFs site, whereas the water oxidation takes place at the CdSe/CdS site.

6. There has been a good number of literature on COF nanospheres and their behaviors. I would recommend the authors to take a look [*J. Am. Chem. Soc.*, 2021, 143, 8426; *J. Am. Chem. Soc.*, 2021, 143, 955; *J. Am. Chem. Soc.*, 2019, 141, 20371; *ACS Nano*, 2021, 15, 12723; *J. Am. Chem. Soc.*, 2021, 143, 20916]

Response: Thanks for the comments. These are interesting papers dealing with COF colloidal particles. We have cited these papers as Ref. 40-44 in the introduction part.

Reviewer #4

The authors in this manuscript fabricated a micro-meso hierarchically porous nanocomposite with CdSe/CdS nanorods and imine-based polymers, which possessed double-shelled hollow structures effective for CO₂-to-CO photoreduction. The hollow structure was characterized in detail. I do have concerns on this manuscript that deter me from recommending its publication, and my questions are as follows:

Response: We thank this reviewer for the comments and suggestions on our present manuscript, and hope our revision will clarify his/her concerns.

1. Structural evidence must be shown for claiming “COF”, since it is not a random name. The authors need to show the crystallinity from X-ray measurements, like GIWAXS, otherwise the organic shell can be only defined as Porous Framework.

Response: Thanks for the comment. We have performed the X-ray diffraction measurements for characterizing the crystalline structure of the sample. Although the crystallinity of the TAPT-DMTA polymer is not in high quality, several peaks in XRD pattern could be observed, identified as (100), (200), and (001) facets of the TAPT-DMTA polymer.

Figure R1. (a) Experimental XRD (black line) profile of TAPT-DMTA-H and the simulated XRD pattern (red line). (b) The simulated unit cell of TAPT-DMTA COFs.

2. In Figure 1, the construction of hollow structures requires fairly harsh conditions with high AcOH concentration up to 60 vol%. While the condition is compatible with TAPT-DMTA-H, I doubt whether such

acidities cause structural, compositional or spectral damage to CdSe/CdS nanorods? Another concern about the acidity is that since hollow structures are formed under acidic conditions, do they remain stable under basic conditions, as CO₂ reduction is anticipated to consume protons and cause local basicity to some extent?

Response: Thanks for the comment. To confirm the stability of the CdSe/CdS NRs after the acidic treatment. We have performed the high-resolution TEM (HRTEM) measurement to identify the lattice structure of the NRs. The lattice fringe of 0.34 nm could be observed, which is assigned to the (0002) crystal plane of the CdS, in agreement with a fast growth rate of the NRs along the *c*-axis of their wurtzite structure (**Figure R2**). Moreover, XRD measurement of the same sample showed that wurtzite phase of CdS could be well observed (**Figure R2b**). Combined the HRTEM and XRD data, we conclude that the CdSe/CdS NRs were not damaged after the acidic treatment.

To further test the stability of the double shelled nanocomposites under basic conditions, the **TAPT-DMTA/NR40-H** sample was dispersed into NaOH aqueous solution (0.1 M) or NaHCO₃ aqueous solution (0.1 M) for 12 hours. These samples were washed with deionized water and were evaluated by TEM and XRD (**Figure R2e-2f**). TEM images showed that the double shelled structures of the nanocomposites were retained after the base treatment. Furthermore, the XRD pattern of the treated samples showed the similar peaks, identified as wurtzite CdS structure. Therefore, we conclude that such nanocomposites are relatively stable against the base treatments.

Figure R2. (a) TEM image of **TAPT-DMTA/NR40-H**; (b) XRD pattern of **TAPT-DMTA/NR40-H**; (c-d) HRTEM images of **TAPT-DMTA/NR40-H**. TEM image (e) and XRD pattern (f) of **TAPT-DMTA/NR40-H** after the treatment in 0.1 M NaHCO₃. TEM image (g) and XRD pattern (h) of **TAPT-DMTA/NR40-H** after the treatment in 0.1 M NaOH.

3. The solid-state UV-Vis absorption spectra of all the samples in Figure 2 show absorption maximum at exactly 1, is it a coincidence? If spectra were normalized, the information has to be mentioned either in the figure or in the caption.

Response: Thanks for pointing out this issue. We have normalized the UV-vis spectra in Figure 2. We have mentioned such normalization treatment in the text. Additionally, we have provided the raw UV-vis spectra without normalization in the Supplementary Information.

Figure R3. The UV-Vis absorption spectra of TAPT-DMTA-H, TAPT-DMTA/NR40-H and TAPT-DMTA/NR40-S without normalization.

4. It was observed that “the exciton lifetime of the TAPT-DMTA/NR40-H was markedly shortened from 63.06 ns of the NR40 to 0.22 ns”. For such a dramatic decline of exciton lifetime, although enhanced charge separation makes contribution, I doubt other reasons can also exist, as in Figure 3c and 3d, the CdSe/CdS NRs are in very tight and condense bundles. In this situation, will self-quenching of adjacent CdSe/CdS NRs also shorten the exciton lifetime? (This self-quenching may not be that helpful to deliver photogenerated electrons for CO₂ reduction)

Response: Thanks for the comment. We agree with this reviewer that aggregation of CdSe/CdS NRs could also lead to the self-quenching of adjacent CdSe/CdS NRs. To identify the effect of aggregation of NRs on their PL lifetime, we measured the PL lifetime of CdSe/CdS assemblies prepared under identical condition without the addition of TAPT/DMTA monomers. The PL lifetime of CdSe/CdS NR assemblies was shortened to 39.96 ns compared to that of the isolated CdSe/CdS NRs (**Figure R4**). Nevertheless, the lifetime of CdSe/CdS NR assemblies is still two orders of magnitude greater than that of the TAPT-DMTA/NR40-H in

the presence of polymers. In the revised manuscript, we used the lifetime of CdSe/CdS NR assemblies as a control to rule out the contribution from self-quenching of adjacent NRs. We thank again this reviewer for pointing out this important issue.

Figure R4. Time resolved PL spectra of (a) NR40, (b) NR40 assemblies, and (c) TAPT-DMTA/NR40-H.

5. The mutual interaction between QDs and COF was only characterized by steady-state and time resolved photoluminescence. I agree with the author that the quenched lifetime indicates the strong interaction between the excited states of QDs and the COF. However, we might be able to distinguish electron transfer or hole transfer. The mechanism as depicted in Figure 4c corresponds to the type II charge separation in heterojunction structure, which should show the slower recombination. Could the author explain a little more about the mechanism?

Response: Thanks for the comment. The 1D CdSe/CdS dot-in-rod is a typical Type-I heterojunction, where the conduction band (CB) and valence band (VB) of CdS NRs straddle the narrower band gap of CdSe core. The CdS NRs act as light harvesting antenna to strongly absorb light and to transport photo excitation energy into the CdSe core where light emission or photochemical reactions can occur. In the present manuscript, we have introduced TAPT-DMTA-COFs, which enables the formation of Type-II heterojunction with CdS, and it could be served as electron acceptors based on the CB energy level of CdS and TAPT-DMTA (**Figure R5a**). Hence, a donor (CdSe)-absorber (CdS)-acceptor/catalyst (TAPT-DMTA) inorganic/organic nanocomposite superstructure is constructed for direct CO₂ photoreduction. Therefore, the distance of charge separation in the charge separated state (CdSe⁺-CdS-TAPT-DMTA⁻) can be simply controlled by the length of CdS rod (**Figure R5b**). Such heterojunction design is akin to the previous report on Pt-tip coated CdSe/CdS dot-in-rod structure, where the CdSe, CdS, and Pt serves as donor, absorber, and acceptor, respectively (Chem. Soc. Rev. 2016, 45, 3781). The distance of charge separation of CdSe/CdS/Pt is strongly correlated with the CdS rod length.

Figure R5. (a) energy level diagram of CdSe, CdS, and TAPT-DMTA in the photocatalytic system on the NHE scale. (b) Representation of the CO₂ photoreduction in the presence of the present CdSe-CdS-TAPT-DMTA heterojunctions. e⁻-h⁺ represents the photo-generated excitonic state in the CdS domain.

6. The authors also performed CO₂ reduction in the absence of sacrificial agents and observed O₂. Does evolved O₂ follow the expected stoichiometry to CO?

Response: Thanks for the comment. We have performed the CO₂ photoreduction without any sacrificial agents. To confirm another half reaction, we have detected the oxygen evolution reaction by GC. First of all, we have conducted a control experiment done without the addition water, and the result showed that no detectable CO could be observed for TAPT-DMTA/NR40-H, indicating that the CO₂ photoreduction cannot proceed in the absence of water. Hence, a most probable half reaction is the water oxidation: $2\text{H}_2\text{O} \rightarrow \text{O}_2 + 4\text{H}^+ + 4\text{e}^-$. We used 50 mg of the TAPT-DMTA/NR40-H photocatalyst to conduct the CO₂ photoreduction, and the products were detected by GC. The standard curve for oxygen gas was first depicted by GC, which was applied to quantify the amount of oxygen generated during the water oxidation reaction. The result in Figure R6 revealed that the stoichiometry between CO and O₂ gas is measured to be around 2:1 at three different reaction times. Hence, we conclude that another half reaction is the oxygen evolution from water oxidation reaction.

Figure R6. (a) Standard curve of the oxygen gas determined by GC. (b) The quantity of CO and O₂ catalyzed by 50 mg of the **TAPT-DMTA/NR40-H** photocatalyst at different reaction times.

7. The data statistics can be only seen in transient photocurrent measurements. However, most of catalytic results, which are the key points of the manuscript, were presented with single measurement. It is very disappointing, given the high expectation on the impact of the manuscript.

Response: Thanks for the comment. We have provided the data statistics for the CO₂ photoreduction experiments. Specifically, we have conducted three independent catalytic experiments for the samples of **NR40** assemblies, **TAPT-DMTA-S**, **TAPT-DMTA-H**, **TAPT-DMTA/NR40-S**, **TAPT-DMTA/NR40-H**, **TAPT-DMTA/NR24-H** and **TAPT-DMTA/NR12-H**. Particularly, the time-dependent CO₂-to-CO performances of **TAPT-DMTA/NR40-H**, **TAPT-DMTA/NR24-H** and **TAPT-DMTA/NR12-H** were conducted. Based on the three independent runs, we evaluated the data statistics. The results showed that the error bar of the CO formation rate for each sample is less than ±10 %, revealing the good reproducibility of the photocatalysts (**Figure R7**).

We have included the data statistics into the revised manuscript. We thank this reviewer again for such a suggestion.

Figure R7. The CO₂-to-CO generation rate of different samples. The samples were measured three runs independently, which were used to determine the error bars in the plot.

8. The quantum yield is an important index to evaluate photocatalytic efficiency. The author should give the catalytic quantum yield of the tests.

Response: Thanks for the suggestion. We have provided the data of external quantum efficiency (EQE) for the sample **TAPT-DMTA/NR40-H**. The EQE is defined as the photocatalytic electron consumption (N_{electron}) to the induced photon flux per hour (N_{photon}) in a given band range, which can be written as follows:

$$\text{EQE (\%)} = N_{\text{electron}}/N_{\text{photon}} = 2N_{(\text{CO})}/N_{\text{photon}} \quad (\text{R1})$$

The calculation of N_{electron} is in close association with the fact that two electrons are consumed to yield one molecule of CO.

The calculation of N_{photon} can be done as follows:

$$N_{\text{photon}} = (I \times A \times t)/E_{\text{photon}} \quad (\text{R2})$$

Where I the light intensity, A is the irradiation area which is fixed to 12.56 cm² in the experiments, and t is the irradiation time that is 10 hours in the experiments. The photon energy $E_{\text{photon}} = hc/\lambda$ (R3), where h is the Planck's constant, c is the speed of light and λ is the specific wavelength of the light using the desired bandpass filter. To calculate the EQE of the sample, we have equipped the light source with a bandpass optical filter (450, 500, 550 or 600 nm) during the photoreduction experiments

Hence, it can be expressed as follows:

$$N_{\text{photon}} = (I \times A \times t) \times \lambda/(hc) \quad (\text{R4})$$

The detailed EQE at each wavelength was calculated and summarized in **Figure R8**.

Wavelength	450 nm	500 nm	550 nm	600 nm
EQE (%)	1.16	0.79	0.64	0.35

Figure R8. The catalytic EQE of **TAPT-DMTA/NR40-H** in the presence of bandpass filter of 450, 500, 550 and 600 nm. The dashed line is the UV-vis absorbance spectrum of **TAPT-DMTA/NR40-H**. The below is the table showing the values of the EQE at various wavelengths.

Reviewer #1 (Remarks to the Author):

The authors revised the manuscript to incorporate the reviewers' comments. I mostly agreed with the authors. Therefore, I recommend publishing this manuscript in this form.

Reviewer #3 (Remarks to the Author):

I think the authors have tackled my and other reviewers' comments satisfactorily. Therefore, I recommend publication.

Reviewer #4 (Remarks to the Author):

In the revised manuscript, the authors addressed most of the reviewer's concerns. However, the active site for CO₂RR is ambiguous although the energy diagram shows the reaction is thermodynamically feasible. With more in-depth understanding of the photocatalytic system, the interactions between CdSe/CdS dot-in-rod and the COF counterpart worth more attention, especially from the atomic or molecular level.

The interaction between CdSe/CdS dot-in-rod and COF was illustrated by steady-state and transient spectroscopy, as well as photocurrent measurement. It is okay to support the macroscopic phenomenon that dot-in-rod nanostructures interact strongly with COF. However, it is unclear how and why dot-in-rod interacts atomically with COF. I read in the rebuttal that their physical contact through van der Waals was the most probable, as there was severe mismatch of lattice constants between dot-in-rod and COF. But even so, based on transient spectroscopy, the PL lifetime reduced dramatically from 39 ns to 0.22 ns, in the presence of COF. It seems that some charge transfer events happen within 10⁻⁹ s, and such rapid charge transfer typically happens when light harvesting units (dot-in-rod structures) and acceptors (COF) have intimate contact directly or just within a few chemical bonds. The mismatch of lattice constants just excludes line-to-line or face-to-face interactions between dot-in-rod and COF over a large distance or area, but some point-to-point interactions between the peripheral COF groups such as (-NH₂) and the dot-in-rod surface may exist. Therefore, the interaction between dot-in-rod and COF, if any, should be better illustrated to have a clear picture for this work.

Other questions need to be addressed

Q1: The authors claimed the small shell thickness of TAPT-DMTA/NR40-H is responsible for the enhanced light absorption of the nanocomposite. Is the thickness of the outer shell and inner shell of the composite material controllable? What is the effect of shell thickness on the photoreduction performance?

Q2: Why use 40 vol% or 60 vol% acetic acid for crystallization? What effect does increasing/decreasing the volume ratio of acetic acid have on the composite?

Q3: The authors claim that several peaks corresponding to the (100), (200) and (001) planes can be observed in the XRD pattern of the TAPT-DMTA polymer, but in the XRD pattern shown in Figure S4e, the peaks corresponding to (200) and (001) are not obvious, especially the (200) crystal plane.

Q4: In the manuscript, the specific surface area is calculated by the Brunauer-Emmett-Teller equation. Detailed calculation process or corresponding reference should be provided.

Q5: Lines 275-277 correspond to Figure 5a and should be annotated.

Q6: When used for comparison, the magnification of the TEM image should be consistent in Figures S3(b) and (d), S27(a) and (b).

Q7: Why is the CdSe/CdS NR length of sample No. 6 in Figure S30 44 instead of 40?

Response to reviewers' comments

Reviewer #1:

The authors revised the manuscript to incorporate the reviewers' comments. I mostly agreed with the authors. Therefore, I recommend publishing this manuscript in this form.

Response: We thank this reviewer for his/her constructive comments, and the opportunity to improve the quality of the manuscript.

Reviewer #2 (from other reviewers):

Based on the referees' evaluations at this stage it is clear that the manuscript was improved upon revision. However, some important points remain to be addressed. In particular, Reviewers #4 has again raised concerns regarding synthetic and characterization details. Further, they found that the response to Reviewer #2's comment #1 was not wholly appropriate and should be revisited. This is in addition to lingering concerns regarding the mechanistic aspects of the work. On balance, their concerns are persuasive enough to request that additional revision and experiments should be performed in order to further verify and elucidate the mechanistic aspects of the results reported in your study.

Response: Thanks for the fruitful suggestion. We have performed additional experiments to confirm the chemical interactions between TAPT-DMTA COFs and CdSe/CdS NRs within the nanocomposites. Following the suggestion from Reviewer #4, the chemical interaction between TAPT-DMTA COFs and CdSe/CdS NRs could be from the peripheral amine group (-NH₂) of TAPT-DMTA COFs and the surface Cd atoms of CdSe/CdS NRs.

It is well known that the primary amine could be capping agents that stabilize CdS or CdSe nanoparticles (J. Am. Chem. Soc. 2003, 125, 11100; Nat. Mater. 2016, 15, 141; Langmuir 2018, 34, 6070;), where these amines are neutral donors that interact with Cd atoms in CdS or CdSe. To confirm the chemical interaction between CdSe/CdS NRs and the peripheral amine group of TAPT-DMTA COF, we first designed an experiment as follows (Route I in **Figure R1**): 7 mg of TAPT molecules was added into 2 mL of THF containing 2 mg of CdSe/CdS NRs (NR40). The optically clear NR colloidal solution became turbid after 8 hours, indicating that these NRs were aggregated in the presence of TAPT molecules that have three terminal amine groups, which would serve as “cross linker” resulting in the flocculation of NRs. To prove that the TAPT molecules are present in the NRs aggregates, infrared (IR) spectroscopy measurements were conducted for the samples, NRs aggregates without the addition of TAPT, TAPT, and NRs aggregates in the presence of TAPT, termed as NRs, TAPT, and NRs/TAPT, respectively (**Figure R2a**). We note that these NR aggregates were separated by centrifugation and washed with THF several times to remove the unbound TAPT molecules. Both the TAPT and NRs/TAPT samples showed IR peaks at ~1500 cm⁻¹ from the C=N stretching of the triazine ring, which is not observed for the NRs in the absence of TAPT, indicating that TAPT molecules are present in the NRs/TAPT sample (**Figure R2a**). We note these NRs/TAPT aggregates could not be dissolved in any common organic solvents, indicating the formation of chemical bonding between TAPT and NRs. Unfortunately, the TAPT molecule within these NRs/TAPT aggregates could not be identified by the ¹H NMR spectroscopy, which is an important tool frequently used to characterize the surface capping agents for colloidal NPs (Chem. Mater. 2013, 25, 1211). To further confirm the chemical bonding between CdSe/CdS NRs and the peripheral amine group of TAPT-DMTA COF, we used aniline as the capping agent to interact with the surface of CdSe/CdS NRs (Route II in **Figure R1**). After adding 7 mg of aniline into 2 mL of THF containing 2 mg of CdSe/CdS NRs, the colloidal solution remained optically clear after 24 hours. These NRs were collected and washed with THF to remove the unbound aniline and were dissolved in CDCl₃ for ¹H NMR measurement. The NMR data in **Figure R2a** clearly showed that NMR signal assigned to the hydrogen from the aromatic ring could be observed in the NRs/aniline sample, indicating that these aniline molecules were chemically bound to the surface of NRs. We note the stabilization of these NRs in solution is mainly from the aliphatic chains from the ODPA/HPA ligands.

Overall, these additional experiments clearly showed that primary amine group (-NH₂) could bound to the surface CdSe/CdS NRs. Hence, we believed that the peripheral amine group of TAPT-DMTA COF could also interact with the surface of CdSe/CdS NRs, which leads to the contact between COF and CdSe/CdS atomically.

This was included in the revised manuscript as Supplementary Note I, and the **Figure R1-R2** were included in the Supplementary Information.

Figure R1. Scheme for additional experiments that confirm the interaction between the amine of COF and the surface of NRs. Two typical routes were applied: Route I and Route II. The native ligands (ODPA/HDA) that capped on the surface of CdSe/CdS NRs were omitted for clarity.

Figure R2. (a) Infrared spectra of NRs, TAPT, and NRs/TAPT aggregates; (b) ^1H NMR spectra of NRs, aniline, and NRs/aniline composites.

Reviewer #3:

I think the authors have tackled my and other reviewers' comments satisfactorily. Therefore, I recommend publication.

Response: We thank this reviewer for his/her constructive comments, and the opportunity to improve the quality of the manuscript.

Reviewer #4:

In the revised manuscript, the authors addressed most of the reviewer's concerns. However, the active site for CO₂RR is ambiguous although the energy diagram shows the reaction is thermodynamically feasible. With more in-depth understanding of the photocatalytic system, the interactions between CdSe/CdS dot-in-rod and the COF counterpart worth more attention, especially from the atomic or molecular level.

The interaction between CdSe/CdS dot-in-rod and COF was illustrated by steady-state and transient spectroscopy, as well as photocurrent measurement. It is okay to support the macroscopic phenomenon that dot-in-rod nanostructures interact strongly with COF. However, it is unclear how and why dot-in-rod interacts atomically with COF. I read in the rebuttal that their physical contact through van der Waals was the most probable, as there was severe mismatch of lattice constants between dot-in-rod and COF. But even so, based on transient spectroscopy, the PL lifetime reduced dramatically from 39 ns to 0.22 ns, in the presence of COF. It seems that some charge transfer events happen within 10⁻⁹ s, and such rapid charge transfer typically happens when light harvesting units (dot-in-rod structures) and acceptors (COF) have intimate contact directly or just within a few chemical bonds. The mismatch of lattice constants just excludes line-to-line or face-to-face interactions between dot-in-rod and COF over a large distance or area, but some point-to-point interactions between the peripheral COF groups such as (-NH₂) and the dot-in-rod surface may exist. Therefore, the interaction between dot-in-rod and COF, if any, should be better illustrated to have a clear picture for this work.

Response: Thanks for pointing out this important issue. In addition to the macroscopic phenomenon that dot-in-rod nanostructures interact strongly with COF, we have provided the additional evidence on the chemical interaction between TAPT-DMTA COF and CdSe/CdS NRs, following the suggestion from this reviewer—the interaction between the peripheral amine group (-NH₂) of TAPT-DMTA COF and CdSe/CdS NRs.

It is well known that the primary amine could be capping agents that stabilize CdS or CdSe nanoparticles (J. Am. Chem. Soc. 2003, 125, 11100; Nat. Mater. 2016, 15, 141; Langmuir 2018, 34, 6070;), where these amines are neutral donors that interact with Cd atoms in CdS or CdSe. To confirm the chemical interaction between CdSe/CdS NRs and the peripheral amine group of TAPT-DMTA COF, we first designed an experiment as follows (Route I in **Figure R1**): 7 mg of TAPT molecules was added into 2 mL of THF containing 2 mg of CdSe/CdS NRs (NR40). The optically clear NR colloidal solution became turbid after 8 hours, indicating that these NRs were aggregated in the presence of TAPT molecules that have three terminal amine groups, which would serve as “cross linker” resulting in the flocculation of NRs. To prove that the TAPT molecules are present in the NRs aggregates, infrared (IR) spectroscopy measurements were conducted for the samples, NRs aggregates without the addition of TAPT, TAPT, and NRs aggregates in the presence of TAPT, termed as NRs, TAPT, and NRs/TAPT, respectively (**Figure R2a**). We note that these NR aggregates were separated by centrifugation and washed with THF several times to remove the unbound TAPT molecules. Both the TAPT and NRs/TAPT samples showed IR peaks at ~1500 cm⁻¹ from the C=N stretching of the triazine ring, which is not observed for the NRs in the absence of TAPT, indicating that TAPT molecules are present in the NRs/TAPT sample (**Figure R2a**). We note these NRs/TAPT aggregates could not be dissolved in any common organic solvents, indicating the formation of chemical bonding between TAPT and NRs. Unfortunately, the TAPT molecule within these NRs/TAPT aggregates could not be identified by the ¹H NMR spectroscopy, which is an important tool frequently used to characterize the surface capping agents for colloidal NPs (Chem. Mater. 2013, 25, 1211). To further confirm the chemical bonding between CdSe/CdS NRs and the peripheral amine group of TAPT-DMTA COF, we used aniline as the capping agent to interact with the surface of CdSe/CdS NRs (Route II in **Figure R1**). After adding 7 mg of aniline into 2 mL of THF containing 2 mg of CdSe/CdS NRs, the colloidal solution remained optically clear after 24 hours. These NRs were collected and washed with THF to remove the unbound aniline and were dissolved in CDCl₃ for ¹H NMR measurement. The NMR data in **Figure R2a** clearly showed that NMR signal assigned to the hydrogen from the aromatic ring could be observed in the NRs/aniline sample, indicating that these aniline molecules were chemically bound to the surface of NRs. We note the stabilization of these NRs in solution is mainly from the aliphatic chains from the ODPA/HPA ligands.

Overall, these additional experiments clearly showed that primary amine group (-NH₂) could bound to the surface CdSe/CdS NRs. Hence, we believed that the peripheral amine group of TAPT-DMTA COF could also interact with the surface of CdSe/CdS NRs, which leads to the contact between COF and CdSe/CdS atomically.

This was included in the revised manuscript as Supplementary Note 1, and the **Figure R1-R2** were included in the Supplementary Information.

Figure R1. Scheme for additional experiments that confirm the interaction between the amine of COF and the surface of NRs. Two typical routes were applied: Route I and Route II. The native ligands (ODPA/HDA) that capped on the surface of CdSe/CdS NRs were omitted for clarity.

Figure R2. (a) Infrared spectra of NRs, TAPT, and NRs/TAPT aggregates; (b) ¹H NMR spectra of NRs, aniline, and NRs/aniline composites.

Other questions need to be addressed

Q1: The authors claimed the small shell thickness of TAPT-DMTA/NR40-H is responsible for the enhanced light absorption of the nanocomposite. Is the thickness of the outer shell and inner shell of the composite material controllable? What is the effect of shell thickness on the photoreduction performance?

Response: Thanks for the comment. Concerning on the thickness of the outer shell and inner shell of the nanocomposites, we have changed the mass ratios between TAPT-DMTA and NR40 during the synthesis. In the manuscript, under a given mass ratio of TAPT-DMTA:NR40 = 12.8:2, double shelled hollow nanocomposites could be observed, and the inner and outer shell thicknesses in **TAPT-DMTA/NR40-H** were determined to be 14.6 ± 3.4 and 29.5 ± 3.5 nm, respectively, by counting more than 200 particles in the TEM images. When the amount of NR40 was decreased (TAPT-DMTA:NR40 = 12.8:1), TEM image in **Figure R3a** showed that the outer shell thickness of the nanocomposites was ~50 nm, which is markedly larger than that of **TAPT-DMTA/NR40-H** (29.5 nm). On the contrary, when the amount of NR40 was increased (TAPT-DMTA:NR40 = 12.8:4), TEM image in **Figure R3b** showed that the nanocomposites were comprised of the single shelled hollow structure comprising the TAPT-DMTA and NR40. Hence, we conclude that the outer shell thickness could be increased to ~50 nm, but it could not be decreased to a thinner shell.

Concerning on the inner shell of the nanocomposites that is comprised of NR40, we found that it is rather difficult to adjust the inner shell thickness. In fact, the inner shell of the nanocomposites is formed from the self-assembly of NRs. Based on the results from spherical nanoparticles, a nanoparticle monolayer could be observed, which is probably formed from the interface-templated self-assembly.

Lastly, we also studied the CO₂ photoreduction performance of the nanocomposites prepared under different ratios between TAPT-DMTA and NR40 (**Figure R3**). For nanocomposites with a thicker outer shell of ~ 50 nm, the CO yield was determined to be 234 $\mu\text{mol g}^{-1}$ after 6 hours, which is markedly lower than that of the nanocomposites with a thinner outer shell (395 $\mu\text{mol g}^{-1}$). Furthermore, for nanocomposites prepared under TAPT-DMTA:NR40 = 12.8:4, the CO yield was determined to be 109 $\mu\text{mol g}^{-1}$ after 6 hours, which is also much lower than that of the double shelled nanocomposites.

We have included these results into the Supplementary Information in rerevised manuscript.

Figure R3. (a) TEM images of nanocomposites prepared under TAPT-DMTA:NR40 = 12.8:1; (b) TEM images of nanocomposites prepared under TAPT-DMTA:NR40 = 12.8:4. (c) plot of the outer shell thickness over the mass ratios. (d) The CO₂-to-CO yield of different samples at different mass ratios (TAPT-DMTA:NR40).

Q2: Why use 40 vol% or 60 vol% acetic acid for crystallization? What effect does increasing/decreasing the volume ratio of acetic acid have on the composite?

Response: Thanks for the comment. We have studied the effect of concentration of acetic acid on the morphology of the nanocomposites. Specifically, the decrease of the volume ratio of acetic acid to 20 vol%, Solid outer shell could be observed (**Figure R4a**). The increase of the volume ratio of acetic acid to 80 vol%, double shelled hollow nanocomposites could be observed (**Figure R4d**), which is similar to that prepared in under 60 vol% of acetic acid in the manuscript.

Figure R4. TEM images of TAPT-DMTA/NR40 nanocomposites prepared under different volume ratio of acetic acid during the crystallization process: (a) 20 vol%; (d) 40 vol%; (c) 60 vol%; (d) 80 vol%.

Q3: The authors claim that several peaks corresponding to the (100), (200) and (001) planes can be observed in the XRD pattern of the TAPT-DMTA polymer, but in the XRD pattern shown in Figure S4e, the peaks corresponding to (200) and (001) are not obvious, especially the (200) crystal plane.

Response: Thanks for the comment. Normally, the crystallinity of the triazine-based organic framework is not very high, which could be found in the reported literature (Angew. Chem. Int. Ed., 2008, 47, 3450; Adv. Mater. 2019, 31, 1807865). The low crystallinity of the triazine-based organic framework is due to the strong aromaticity of the triazine ring, which results in the formation of imine bonds easily during the synthesis of triazine-based organic framework, thereby hampering the formation of ordered structures during the crystallization process.

Q4: In the manuscript, the specific surface area is calculated by the Brunauer-Emmett-Teller equation. Detailed calculation process or corresponding reference should be provided.

Response: Thanks for comment. We have cited the reference regarding to the surface area calculated by the Brunauer-Emmett-Teller (BET) equation (J. Am. Chem. Soc. 1938, 60, 309-319).

Q5: Lines 275-277 correspond to Figure 5a and should be annotated.

Response: Thanks for the suggestion. It has been done in the revise manuscript.

Q6: When used for comparison, the magnification of the TEM image should be consistent in Figures S3(b) and (d), S27(a) and (b).

Response: Thanks for the comment. This issue has been fixed in the revised manuscript. Figure S27 showed the TEM images of the same sample of two different magnifications.

Q7: Why is the CdSe/CdS NR length of sample No. 6 in Figure S30 44 instead of 40?

Response: Thanks for pointing out this issue. It was a mistake, and we have corrected it in the revised manuscript.

Reviewer #4 (Remarks to the Author):

In the revised manuscript, the authors addressed my concerns for the interaction between COF and nanorods, as well as questions from other reviewers. I think the manuscript can be published in current form.

Response to reviewers' comments

Reviewer #4 (Remarks to the Author):

In the revised manuscript, the authors addressed my concerns for the interaction between COF and nanorods, as well as questions from other reviewers. I think the manuscript can be published in current form.

Response: We thank the reviewer for the valuable comments and we are glad that the reviewer is satisfied with our revisions.